# BROWSENET: GRAPH-BASED ASSOCIATIVE MEMORY FOR CONTEXTUAL INFORMATION RETRIEVAL

**Pavan Kumar S**[*], **Kiran Kumar Nakka**[*], **C Vamshi Krishna Reddy, Nirav P Bhatt** [†]
Department of Data Science and Artificial Intelligence
Indian Institute of Technology Madras, India
{spavaniitm, me20b117, ch20b112, niravbhatt}@smail.iitm.ac.in

**Divyateja Pasupuleti, Prakhar Agarwal, Harpinder Jot Singh, Anshu Avinash**
DevRev, India
{divyateja.pasupuleti, prakhar.agarwal, harpinder.singh, anshu.avinash}@devrev.ai

## ABSTRACT

Associative memory systems face significant challenges in efficiently retrieving semantically related information from large document collections, particularly when queries require traversing complex relationships between concepts. Traditional retrieval-augmented generation (RAG) approaches often struggle to capture intricate associative patterns and relationships embedded within textual data. To address this limitation, we propose BrowseNet, a novel associative memory framework that leverages query-specific subgraph exploration within a named-entity-based graph for enhanced information retrieval. Our method transforms unstructured text into a graph-of-chunks representation, where nodes encode document chunks with semantic embeddings and edges capture lexical relationships between content segments. By dynamically traversing the graph-of-chunks based on query characteristics, BrowseNet emulates content-addressable memory systems that enable efficient pattern matching and associative recall. The framework incorporates both structural similarity derived from lexical relationships and semantic similarity based on embedding representations to optimize retrieval performance. We evaluate BrowseNet against established RAG baselines and state-of-the-art (SOTA) pipelines using publicly available datasets that require associative reasoning across multiple information sources. Experimental results demonstrate that BrowseNet achieves SOTA performance in exact match score over both the graph-based RAG approaches and the dense retrieval methods. The two-pronged approach combining structural graph traversal with semantic embeddings enables more effective associative memory retrieval, particularly for queries requiring the integration of disparate but related information. The code and datasets are open-sourced at: https://github.com/bisect-group/BrowseNet

## 1 INTRODUCTION

The emergence of Large Language Models (LLMs) such as GPT-5 (OpenAI., 2025b; Achiam et al., 2023), Gemini (Google., 2025; Reid et al., 2024), and LLaMA (Dubey et al., 2024) has marked a transformative era in artificial intelligence. These models excel at pattern recognition and coherent text generation, but face inherent limitations in incorporating new information without costly retraining. This challenge has motivated research into frameworks that allow LLMs to update and associate knowledge in more flexible, human-like ways. Retrieval-Augmented Generation (RAG) (Lewis et al., 2020; Gao et al., 2023) represents a key advance in this direction. By separating knowledge storage from reasoning, RAG systems enable models to access external repositories that can be updated independently of the model parameters. This modular design improves adaptability and scalability, yet current RAG pipelines often focus on retrieving documents based on literal similarity, overlooking

---

[*]These authors contributed equally to this work.
[†]Corresponding author

richer associative relationships. As a result, they struggle with queries that require reasoning across multiple contexts or integrating disparate pieces of information.

Associative memory offers a natural solution to this limitation. In human cognition, associations between concepts enable flexible recall that extends beyond surface similarity, allowing connections across contexts and domains (Suzuki, 2005; Yao et al., 2023). These bidirectional associations form semantic networks that facilitate knowledge transfer and generalization, capabilities that remain challenging for current AI systems. Incorporating such associative mechanisms into retrieval systems is therefore a crucial step toward more robust memory architectures for LLMs (Wu et al., 2025). Multi-hop question answering (MHQA) exemplifies this challenge. Standard RAG approaches typically retrieve isolated chunks without modeling their interconnections, which limits their effectiveness for multi-step reasoning. To compensate, existing methods rely on iterative prompting strategies that require multiple interactions with LLMs (Trivedi et al., 2022a; Wei et al., 2022; Yao et al., 2023; Wang et al., 2024). While effective, these approaches increase latency and inference costs (Gutiérrez et al., 2024).

To address these gaps, we propose BrowseNet, a graph–based associative memory framework that unifies lexical and semantic retrieval approaches. BrowseNet transforms unstructured text into a lexically connected graph-of-chunks, where edges capture entity co-occurrence and syntactic relations, and nodes are enriched with semantic embeddings. By framing MHQA as a query-specific graph traversal problem, BrowseNet constructs query-subgraphs that reflect the structural and semantic requirements of complex questions (refer to Fig. 1-(a)). This approach enables the system to link decomposed single-hop queries into coherent reasoning chains, retrieving information more efficiently and effectively than conventional retrievers.

The contributions of this work are as follows: (1) BrowseNet dynamically adapts traversal paths through the graph-of-chunks according to the query's structural and semantic features, (2) The framework integrates lexical and semantic relationships, allowing more nuanced and context-aware retrieval than single-modality methods, (3) Retrieval is achieved with a single LLM interaction, guided by pre-generated decomposed queries, reducing cost and latency, and (4) Empirical evaluation shows that BrowseNet achieves state-of-the-art performance in multi-hop question answering, outperforming both dense retrievers and graph-based RAG systems.

## 2    RELATED WORKS

**Retrieval Augmented generation (RAG):**  RAG was introduced by Lewis et al. (2020) to overcome the limitations of traditional fine-tuning for LLMs. By enabling dynamic access to external knowledge, RAG improves factual accuracy and adaptability. Its framework comprises three stages: Indexing, where documents or document chunks are encoded and stored in a vector database; Retrieval, where relevant text chunks are fetched; and Generation, where the LLM produces responses using an augmented prompt Gao et al. (2023). BrowseNet departs from this pipeline at every stage. In indexing, it constructs a graph-of-chunks that integrates both lexical relationships and semantic embeddings. In retrieval, instead of isolated chunks, BrowseNet extracts a query-specific subgraph that preserves reasoning dependencies among information units. Finally, in generation, the augmented prompt incorporates decomposed sub-queries, enabling the LLM to perform structured reasoning over the retrieved content.

**Graph Informed Retrieval Augmented Generation:**    Naive RAG pipelines often fall short when queries require integrating information across multiple documents. To address this, several graph-based extensions have been proposed. GraphRAG (Edge et al., 2024) constructs hierarchical knowledge graphs (KG) to improve reasoning over complex relationships. RAPTOR (Sarthi et al., 2024) employs recursive clustering and summarization to build tree-structured document representations that support multi-level retrieval. LightRAG (Guo et al., 2024) enhances text indexing with graph structures and introduces a dual-level retrieval mechanism for greater efficiency and contextual accuracy. While these methods improve cross-context reasoning, they rely heavily on LLMs during indexing to generate or expand the retrieval corpus. This reliance increases costs and introduces noise from LLM-generated text. In contrast, BrowseNet minimizes such dependence: generative-LLMs are used only for graph link inference during indexing and for query decomposition during retrieval. Our best-performing setup requires no LLM involvement in offline indexing, making it both cost-efficient and less prone to noise.

**Brain-Inspired RAG:** HippoRAG (Gutiérrez et al., 2024) and HippoRAG 2 (Gutiérrez et al., 2025) advance RAG by introducing brain-inspired mechanisms that emulate associative memory for improved integration of retrieved knowledge. HippoRAG 2 currently achieves state-of-the-art performance in MHQA. However, these approaches require both named entity recognition (NER) and relation extraction (RE) for KG construction. In contrast, BrowseNet requires only NER, simplifying the pipeline while maintaining high retrieval effectiveness.

## 3 METHODOLOGY

The overall workflow of the proposed approach, BrowseNet, is illustrated in Fig. 1-(b). It is structured into three phases: (1) Graph-of-chunks construction, (2) Context retrieval, and (3) Answer generation.

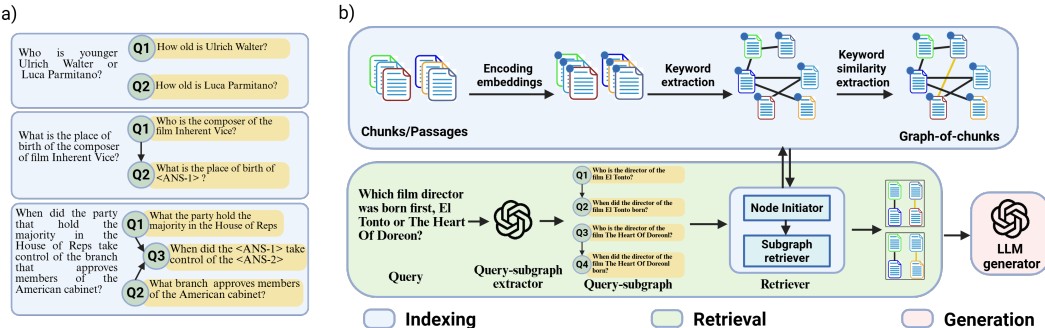

Figure 1: a) Decomposition of the multi-hop query: The multi-hop queries are decomposed into single-hop queries, structurally linked according to answer dependencies. b) Overview of the BrowseNet workflow: The indexing phase includes chunking, embedding encoding, and keyword extraction to construct the graph-of-chunks. During retrieval, BrowseNet employs LLMs to extract the query-subgraph and identifies structurally similar and semantically relevant subgraphs within the graph-of-chunks. Finally, the generation phase leverages the retrieved subgraphs and queries to generate answers.

### 3.1 GRAPH-OF-CHUNKS CONSTRUCTION

Let $G = (V, E)$ be a graph-of-chunks constructed from a corpus of documents $D$, where $V$ denotes a set of nodes corresponding to passages (or chunks) of documents, and $E$ represents the set of edges between these nodes. Each node $c \in V$ is associated with three key attributes: a unique index that identifies the passage, the text of the passage along with its title, and a semantic vector that captures its contextual meaning. The NV-Embed-v2 model (Lee et al., 2024) is used to encode the entire corpus into vector embeddings. Let $M(c)$ denote the embedding of the chunk $c$. Also, ablations are done on the encoders, GTE-Qwen2 (7B) (Li et al., 2023) and Granite-125M-English (Awasthy et al., 2025) (refer to Section 5.4). An edge $e_{ij} \in E$ exists between two nodes $(c_i, c_j \in V)$ if they share a common or synonymous entity, thereby capturing lexical relationships between passages. The construction of the graph-of-chunks involves two main steps: **1) Named Entity Recognition (NER):** Identifying named entities within each passage, and **2) Entity Linking:** Detecting synonymous or semantically related entities to establish additional edges.

**NER** is performed using GLiNER (a BERT-based model) (Zaratiana et al., 2023), which supports zero-shot entity extraction based on provided label sets. We observed that results are largely consistent when using generative models based on GPT-like architectures (e.g., GPT-4o (OpenAI., 2025a) and Claude-3.7-Sonnet (Anthropic., 2024)) in place of GLiNER. For GLiNER, NER requires pre-defined labels as input. Given that the benchmark datasets used are general-purpose rather than domain-specific, we adopt broad category labels such as event, facilities, language, location, money, nationality, religious, political, organization, person, product, work_of_art, occupation, time, ordinal, and date. For GPT-like models, a one-shot demonstration is used for entity extraction (refer Appendix A.1 for prompts). All extracted entities are post-processed to retain only alphanumeric

characters and spaces. **Entity linking** is performed using the ColBERTv2 model (Santhanam et al., 2021) to identify synonymous entities (e.g., TV host and TV presenter). Entities extracted from the NER step are input into ColBERTv2 to compute pairwise similarity scores. Numerical entities and dates are excluded from this step as they are less informative for establishing semantic relationships. Pairs of entities with a cosine similarity score greater than 0.9 are considered synonymous (refer to Section 5.4 for ablation studies on other thresholds). This allows passages that contain equivalent entities to be connected in the graph, enriching the structure. A snippet of the graph-of-chunks with seven nodes from the 2WikiMultiHopQA corpus is shown in Fig. 2 in Appendix.

## 3.2 CONTEXT RETRIEVAL

Context retrieval for a given question consists of two essential steps: 1) Query-subgraph extraction, and 2) Graph-of-chunks traversal.

### 3.2.1 QUERY-SUBGRAPH EXTRACTION

Each multi-hop query, $Q_{orig}$ can be decomposed into a sequence of single-hop queries, where each single-hop query builds upon the answer to the previous one (Fig. 1). We model this multi-hop query as a directed graph, referred to as the *query-subgraph*, where nodes correspond to individual single-hop queries, and directed edges represent the dependency between them, linking queries through their intermediate answers. This directed subgraph has to be *acyclic* and can have more than one connected component (Example question used in Fig. 1-(b) has two connected component), reflecting the inherent structure of multi-hop question answering. Circular dependencies would imply that a subquestion requires its own answer as a prerequisite, leading to ill-defined and non-terminating reasoning. Consistent with this design choice, all gold query decompositions in standard benchmarks (HotpotQA, 2WikiMQA, MuSiQue) are acyclic. Restricting decomposition to Directed Acyclic Graphs (DAGs), therefore, ensures semantic validity, guaranteed termination, and tractable reasoning in practical multi-hop QA settings. We have employed the GPT-4o model (OpenAI., 2025a) to generate the query-subgraph (refer Appendix A.1 for prompts). Furthermore, ablations are done on DeepSeek Reasoner (Guo et al., 2025), GPT-4o-mini (OpenAI., 2025c) and Claude-3.7-Sonnet (Anthropic., 2024).

### 3.2.2 GRAPH-OF-CHUNKS TRAVERSAL FOR CONTEXT RETRIEVAL

Once the query-subgraph is identified, the retrieval process involves traversing the Graph-of-chunks $\boldsymbol{G}$ to extract a subset of chunks from $\boldsymbol{V}$ that provide contextual information for answer generation of the query $Q_{orig}$. Formally, let $\boldsymbol{Q} = (\boldsymbol{V_q}, \boldsymbol{E_q})$ denote the query-subgraph, where $\boldsymbol{V_q}$ represents a set of nodes corresponding to the decomposed single-hop queries, and $\boldsymbol{E_q}$ denotes the set of directed edges capturing dependencies between them. The retrieval begins by identifying the connected components, $Q^{(i)} \in \boldsymbol{Q}$, each of which is processed separately for subgraph extraction. Within each connected component, nodes are sorted in topological order, with initiator nodes (those with no incoming edges) at the beginning and terminal nodes (those with no outgoing edges) at the end. Consider the example query shown in Fig. 1-(b), $Q1$ and $Q3$ are the initiator nodes, and $Q2$ and $Q4$ are the terminal nodes. Retrieval is performed for each node in the order of the topological sort to extract its respective candidate chunks. Our approach accounts for noise and errors in query decomposition and lexical relationship extraction, and accordingly designs a robust retrieval algorithm. Two distinct approaches are employed for initiator nodes and non-initiator nodes.

**Retrieval for Initiator Nodes:** For each initiator node, the top-$k$ candidate chunks (denoted $c_1, \ldots, c_k \in \boldsymbol{V}$) with the highest similarity scores are retrieved by treating all nodes (chunks) in the graph-of-chunks as the corpus. The similarity score for a chunk $c_i$, denoted $SS_{c_i}$, is computed as the maximum cosine similarity between its embedding ($\boldsymbol{M}(c_i)$), the embeddings of the original multi-hop query ($\boldsymbol{M}(Q_{orig})$) and the corresponding single-hop query ($\boldsymbol{M}(V_q^j)$), using Equation 1:

$$SS_{c_i} = max(cos\angle(\boldsymbol{M}(c_i), \boldsymbol{M}(Q_{orig})), cos\angle(\boldsymbol{M}(c_i), \boldsymbol{M}(V_q^j))) \tag{1}$$

This approach is designed to mitigate the effects of noise or errors introduced by incorrect query decomposition. The intuition is that if a single-hop query accurately captures the needed information, the corresponding chunk will exhibit higher similarity to it than to the multi-hop query. Conversely, if the single-hop query is poorly formulated by the LLM, the original multi-hop query may still retrieve

the relevant chunk effectively. Also, note that when a query cannot be decomposed into subqueries, the retrieval procedure falls back to the single initiator node case, which is semantic search over the entire corpus.

**Retrieval for Non-Initiator Nodes:** For a non-initiator node with $p$ predecessor nodes, $P = \{V_q^j : V_q^j$ is a predecessor of $V_q^i\}$ in the query subgraph (retrieval proceeds in topological order), each predecessor has $k$ retrieved candidate chunks. All possible combinations formed by selecting one chunk from each predecessor's retrieved set are considered, resulting in a total of $k^p$ combinations. For each such combination, the candidate chunks for the non-initiator node are defined as the union of the neighbors of the selected predecessor chunks in the graph-of-chunks. This approach is based on the assumption that the graph-of-chunks captures meaningful relationships between the single-hop queries of the predecessor nodes and the target non-initiator node. By considering neighbors in the graph-of-chunks, the method leverages structural information to guide multi-hop retrieval. The implementation details are follows.

A modified query is created by concatenating a chunk from the combination to the current single-hop query, if the query is semantically more similar to the chunk than to any of the neighbors. Subsequently, a semantic search is conducted by assigning each chunk in the neighbors of each combination a similarity score, defined as the maximum of its cosine similarity with the current single-hop query, the modified query, and the original multi-hop query. Now, for each combination, the top-$k$ chunks with the highest scores are selected, resulting in $k^{(p+1)}$ subgraphs. Each subgraph is then scored using a weighted average of the similarity scores of its chunks, where the weight for each chunk is the inverse of the index of the subquery in the topological order, as shown in Equation 2

$$weight_{SG} = \sum_i \frac{SS_{c_i}}{depth_{c_i}} \tag{2}$$

Where $depth_{c_i}$ refers to the minimum depth of the corresponding subquery from an initiator node. where $i$ refers to the index of the subquery in the topological order. The weighting scheme places greater emphasis on initial nodes compared to later ones, as incorrect retrieval at initial nodes can result in entirely erroneous neighbors and, consequently, incorrect subgraph retrieval. The top-$k$ subgraphs with the highest scores are retained, and the corresponding chunks for the current node within these subgraphs are selected. This approach is analogous to beam search applied over subgraphs, where a fixed number of high-scoring candidate subgraphs are maintained (top-$k$), enabling efficient exploration of multiple reasoning hypotheses while focusing retrieval on the most promising paths. The example retrieval for a query with two-hops is shown in Appendix Fig. 3

Although the theoretical space of possible subgraphs is $k^{p+1}$, this upper bound is rarely realized in practice. Realistic query structures are shallow, typically with at most four predecessors ($p \leq 4$), and even $p = 4$ is already uncommon. With a practical choice of $k = 5$, the full combinatorial space would contain at most $5^5 = 3125$ candidate subgraphs. However, because only the top-$k$ subgraphs are retained at each expansion step, the effective search complexity is drastically reduced, making the approach computationally feasible.

**Context Curation:** Once retrieval is performed for every node in the query subgraph following the topological order, the context is formed by the chunks from the top-$k$ subgraphs retained at the end of the retrieval process. Here, $k$ is a hyperparameter referred to as $n\_subgraphs$ hereafter. The structured algorithm, along with its pseudo-code, for traversing the graph-of-chunks to retrieve the most relevant subgraphs is presented in Algorithm 1 in the Appendix.

### 3.3 ANSWER GENERATION

Once the relevant context is retrieved for a given question, it is provided as input to the LLM. Along with the retrieved context, we incorporate an instruction prompt. This prompt guides the model in generating a well-structured response using the generated subqueries that provide the final answer and includes a detailed explanation of the reasoning process that led to the derived conclusion. This ensures that the model's response is both interpretable and transparent, thereby enhancing the system's overall reliability in knowledge-intensive tasks, which enables the generation of an answer along with the reasoning that led to the answer. The prompt used is provided in the Appendix A.1. To align with the previous methods, the LLM used to generate the answer is gpt-4o-mini. However, ablations are done for answer generation using GPT-3.5-turbo, GPT-4.1-mini, Deepseek-Chat-C3, and Gemini-2.0-Flash.

## 4 EXPERIMENTAL SETUP

### 4.1 DATASETS

We evaluate our approach on three benchmark multi-hop question answering datasets: HotpotQA (Yang et al., 2018), 2WikiMQA (Ho et al., 2020), and MuSiQue (Trivedi et al., 2022b). For each dataset, we randomly sample 1,000 questions from the validation split to construct the set of queries and the associated corpus, as performed previously (Press et al., 2022; Trivedi et al., 2022a; Gutiérrez et al., 2024). To better reflect real-world use cases, we have modified the benchmark datasets by including all passages from other questions as candidate distractors. The number of passages for the benchmarks are 9,221, 6,119 and 11,656 respectively for HotpotQA, 2WikiMQA and Musique. This effectively enlarges the candidate corpus, simulating a more realistic retrieval setting where numerous irrelevant documents must be filtered.

### 4.2 BASELINES

We compare BrowseNet with a range of retriever-based approaches, including: (i) simple retrievers such as BM25 (Robertson & Walker, 1994), Contriever (Izacard et al., 2021), and GTR (Ni et al., 2021); (ii) dense retrievers such as NV-Embed-v2 (Lee et al., 2024), GTE-Qwen2 (Li et al., 2023), Granite-125M-English (Awasthy et al., 2025), and Proposition (Chen et al., 2024); and (iii) Graph-augmented RAG methods, including RAPTOR (Sarthi et al., 2024), GraphRAG (Edge et al., 2024), LightRAG (Guo et al., 2024), HippoRAG (Gutiérrez et al., 2024), SiReRAG (Zhang et al., 2024), and HippoRAG-2 (Gutiérrez et al., 2025).

To ensure a fair comparison across all baselines, we employ the same LLM, gpt-4o-mini, for all stages that rely on a generative-LLM, including indexing, retrieval, and question answering. This applies to methods such as RAPTOR, HippoRAG, HippoRAG-2, LightRAG, GraphRAG, and SiReRAG. HippoRAG-2 additionally requires an embedding model, for which we use NV-Embed-v2, the same embedding model employed in BrowseNet. For BrowseNet, we conduct extensive experiments with multiple LLMs and report the best performance obtained. Ablation studies further demonstrate that BrowseNet remains robust across different choices of generative-LLMs.

### 4.3 EVALUATIONS

BrowseNet is evaluated at three stages: graph-of-chunks construction, context retrieval, and answer generation.

**Graph-of-chunks Evaluation:** The constructed graph-of-chunks is evaluated based on its ability to capture the edges necessary to answer multi-hop queries. These required edges are derived from the gold evidence paths provided in the benchmark datasets (refer Appendix A.2). Specifically, MuSiQue and 2WikiMQA include annotated reasoning paths that link the chunks involved in answering each question. The quality of the constructed graph-of-chunks is assessed by measuring the fraction of its edges that correctly represent these gold reasoning paths, thereby reflecting its effectiveness in supporting multi-hop reasoning.

**Query-Subgraph Evaluation:** To evaluate the quality of the generated query-subgraph, we define a metric called isomorphic accuracy (refer Appendix A.3), which captures the structural similarity between the generated subgraph and the gold reasoning pathway provided in the 2WikiMQA and MuSiQue datasets. Two graphs, $G_1$ and $G_2$, are considered isomorphic if there exists a bijective function $f$ that maps the vertices of $G_1$ to the vertices of $G_2$, such that adjacency is preserved. In other words, an edge exists between vertices $u$ and $v$ in $G_1$ if and only if an edge exists between $f(u)$ and $f(v)$ in $G_2$. In our setting, $G_1$ corresponds to the generated query-subgraph, and $G_2$ is the gold graph derived from the annotated reasoning path or query decomposition provided in the benchmark datasets. Isomorphism checking is carried out using NetworkX's $is\_isomorphic()$ function, which implements the exact VF2 algorithm (Cordella et al., 2001). While graph isomorphism is hard in general, the query-subgraphs in our evaluation are very small (maximum four nodes), making exact isomorphism testing computationally feasible across all benchmarks. The results for isomorphic accuracy are presented in the Appendix A.3.

**Context Retrieval Evaluation:** We evaluate context retrieval performance using the *Recall@k* metric. For each question, *Recall@k* is defined as: $\text{R@k} = \frac{|\text{Top-}k \text{ Retrieved Passages} \cap \text{Gold Passages}|}{|\text{Gold Passages}|}$. Here, Gold Passages refer to the set of chunks required to answer the given multi-hop query, as specified

in the benchmark datasets. The final Recall@k score is obtained by averaging the recall across all questions in the evaluation set.

**Answer Generation Evaluation:** We evaluate the quality of generated answers using two standard metrics: Exact Match (EM) and F1 score.

**Exact Match (EM):** The Exact Match metric measures whether the generated answer matches the ground truth answer exactly, word for word. It returns a score of 1.0 for an exact match and 0.0 otherwise. The final EM score reported in the results section is the average of EM scores across all evaluation questions.

**F1 Score:** The F1 score evaluates the overlap between the generated and ground truth answers at the token level. Both answers are tokenized by splitting on whitespace. Precision is defined as the ratio of overlapping tokens to the total number of tokens in the generated answer, while Recall is the ratio of overlapping tokens to the total number of tokens in the ground truth answer. The F1 score is then computed as the harmonic mean of precision and recall: $F1 = 2 \times \frac{\text{Precision} \times \text{Recall}}{\text{Precision} + \text{Recall}}$. The final F1 score is reported as the average of the overall questions in the evaluation set.

## 5 RESULTS AND DISCUSSIONS

### 5.1 EVALUATION OF GRAPH-OF-CHUNKS

The graph-of-chunks constructed for BrowseNet are evaluated based on their ability to retrieve the relevant edges necessary for answering multi-hop queries. This assesses the effectiveness of the graph-of-chunks in identifying key relationships and entities in the corpus. Table 1 shows that, in the 2WikiMQA dataset, the graph-of-chunks achieves an edge accuracy nearing 99.86%, indicating that nearly all essential subgraphs required for reasoning are successfully captured. In contrast, for the MuSiQue dataset, the GLiNER model achieves an edge accuracy of 91.03%. Additionally, the number of entities extracted remains almost consistent across the different NER models used in the study (refer Table 14 in Appendix). The synonymity threshold in ColBERT, used for graph-of-chunks construction, has minimal effect on edge accuracy for the 2WikiMQA dataset across all NER models, as shown in Table 11 in the Appendix. However, for the MuSiQue dataset, edge accuracy is sensitive to the threshold used. The smaller the threshold, the greater the graph's density (refer Appendix A.4), which introduces noise edges (false positives). Also, the ablation studies on ColBERT synonymity threshold (refer to Table 4) show little to no variation in the retrieval. Hence, a larger threshold is chosen to reduce the latency period. We also evaluate retrieval and latency improvements on using a graph-of-chunks in the Appendix A.5. Results in Table 8 in Appendix A.5 shows that Recall@5 improves by approximately 5% compared to the baseline (BrowseNet without graph-of-chunks), while the computation time is reduced by about 1.5 times compared to the baseline. These results underscore the importance of a well-constructed graph-of-chunks in enhancing both the accuracy and efficiency of multi-hop question answering systems.

Table 1: Graph-of-chunks statistics and performance across different datasets.

| NER Model | Dataset | No. of Nodes | No. of Entities | Graph Density | Edge Accuracy |
|---|---|---|---|---|---|
| | HotPotQA | 9,221 | 60,862 | 0.0641 | NA |
| GLiNER | 2WikiMQA | 6,119 | 44,907 | 0.0978 | 99.86 |
| | MuSiQue | 11,656 | 67,332 | 0.0498 | 91.03 |

### 5.2 RETRIEVAL RESULTS

The retrieval performance of BrowseNet is assessed against state-of-the-art (SOTA) models and baseline methods. GraphRAG, SiReRAG, and LightRAG do not follow the retrieve-and-read paradigm; hence, their results are not included in this comparison. Retrieval effectiveness, measured by Recall@2 (R@2) and Recall@5 (R@5), demonstrates that BrowseNet achieves the highest average performance across the three benchmark datasets, as shown in Table 2, thereby establishing new SOTA results, while HippoRAG-2 is the second-best performing pipeline.

In the HotpotQA dataset, the NV-Embed-v2 retriever achieves slightly better performance at R@2. Although HotpotQA requires two-hop reasoning, prior studies (Gutiérrez et al., 2024; Trivedi et al., 2022b) have identified it as a weaker benchmark for multi-hop retrieval due to the presence of

spurious signals. Nevertheless, BrowseNet outperforms all baselines at R@5 in this dataset. In the 2WikiMQA dataset, the query subgraphs typically contain connected components of maximum length two, allowing keywords alone to serve as effective linking mechanisms between passages. Here, BrowseNet surpasses HippoRAG-2 (the previous SOTA) by 2% in both R@2 and R@5. Conversely, the MuSiQue dataset presents a more challenging setting, with query-subgraph components extending up to four hops. This necessitates traversing multiple connections to retrieve relevant context. While NV-Embed-v2 relies solely on semantic similarity, BrowseNet integrates both keyword-based linking and semantic proximity. This hybrid strategy improves its performance in complex multi-hop scenarios, enabling it to capture both explicit and implicit relationships between information sources. As a result, BrowseNet outperforms HippoRAG-2 in recall evaluations. Latency evaluations (refer Appendix A.6) indicate that BrowseNet exhibits slightly higher latency than HippoRAG-2. Nonetheless, the increase is marginal (0.49 seconds on average) and is justified by the substantial improvements in retrieval recall.

Table 2: BrowseNet pipeline outperforms other baselines in retrieval performance in all three benchmarks tested. The best score is written in **Bold** and the second best is underlined. A detailed statistical analysis of closely competing metrics is provided in Appendix A.11.

| Retriever | HotpotQA | | 2WikiMQA | | MuSiQue | | Average | |
|---|---|---|---|---|---|---|---|---|
| | R@2 | R@5 | R@2 | R@5 | R@2 | R@5 | R@2 | R@5 |
| **Simple baselines** | | | | | | | | |
| BM25 | 55.40 | 72.20 | 51.80 | 61.90 | 32.20 | 41.20 | 46.50 | 58.43 |
| Contriever | 57.20 | 75.50 | 46.60 | 57.50 | 34.80 | 46.60 | 46.20 | 59.87 |
| GTR | 59.40 | 73.30 | 60.20 | 67.90 | 37.40 | 49.10 | 52.33 | 63.43 |
| **Dense retrievers** | | | | | | | | |
| NV-Embed-v2 (7B) | **83.95** | 95.65 | 69.05 | 76.72 | 53.30 | 69.85 | 68.77 | 80.74 |
| GTE-Qwen2 (7B) | 72.70 | 87.50 | 65.22 | 73.25 | 48.78 | 63.45 | 62.23 | 74.73 |
| Granite-125M-English | 70.40 | 85.30 | 64.12 | 70.62 | 44.54 | 59.37 | 59.69 | 71.76 |
| Proposition | 58.70 | 71.10 | 56.40 | 63.10 | 37.60 | 49.30 | 50.90 | 61.17 |
| **KG-augmented RAGs** | | | | | | | | |
| RAPTOR | 58.10 | 71.20 | 46.30 | 53.80 | 35.70 | 45.30 | 46.70 | 56.77 |
| HippoRAG | 60.05 | 78.10 | 70.40 | 87.87 | 41.86 | 53.57 | 57.44 | 73.11 |
| HippoRAG-2 | 81.80 | 96.20 | 74.60 | 90.20 | 53.50 | **74.20** | 69.97 | 86.87 |
| BrowseNet | 83.85 | **96.40** | **76.77** | **93.30** | **55.12** | 73.91 | **71.91** | **87.87** |

Table 3: BrowseNet pipeline outperforms other baselines in answer generation in all three benchmarks tested. The best score is written in **Bold** and the second best is underlined. BrowseNet's performance over the second best method is statistically significant based on a paired bootstrap test ($p < 0.05$).

| Retriever | HotpotQA | | 2WikiMQA | | MuSiQue | | Average | |
|---|---|---|---|---|---|---|---|---|
| | EM | F1 | EM | F1 | EM | F1 | EM | F1 |
| NV-Embed-v2 (7B) | 59.80 | 75.52 | 52.50 | 62.57 | 36.90 | 49.80 | 49.73 | 62.63 |
| LightRAG | 9.90 | 20.20 | 2.50 | 12.10 | 2.00 | 9.30 | 4.8 | 13.87 |
| GraphRAG | 51.40 | 67.60 | 45.70 | 61.00 | 27.00 | 42.00 | 41.37 | 56.87 |
| SiReRAG | 48.30 | 63.17 | 41.30 | 48.05 | 26.00 | 39.59 | 38.53 | 50.27 |
| HippoRAG-2 | 59.30 | 76.90 | 60.50 | 69.70 | 35.00 | 49.30 | 51.60 | 65.30 |
| BrowseNet | **62.20** | **77.69** | **63.90** | **74.50** | **41.60** | **54.08** | **55.90** | **68.76** |

## 5.3 QUESTION ANSWERING RESULTS

In the QA evaluation, BrowseNet demonstrates superior performance over all other methods across all benchmarks, achieving higher exact match and F1 scores as reported in Table 3. The performance gains can be attributed to the inclusion of sub-queries in the prompt alongside the original question,

enabling the language model to perform intermediate reasoning steps that reduce ambiguity and enhance contextual understanding. Furthermore, the overall LLM cost of HippoRAG-2 is $33\times$ higher than that of the BrowseNet pipeline, from indexing through to question-answering (refer Appendix A.7).

## 5.4 ABLATIONS STUDIES

We performed ablations at three stages of answer generation: Graph-of-chunks construction, retrieval, and answer generation to understand the contribution and importance of different components in BrowseNet. Results for retrieval ablations are shown in Table 4, with the others in Appendix A.8. BrowseNet shows stable performance across synonymity thresholds and keyword generation models. Increasing the number of subgraphs ($n\_subgraphs$) improves context retrieval but increases latency, making larger values preferable when context size is not limiting and latency is less critical. Retrieval effectiveness was consistent across query decomposition models (DeepSeek Reasoner and gpt-4o-mini), indicating robustness to this choice. On the other hand, a significant drop in performance was observed when BrowseNet was evaluated with a different encoder, underscoring the critical role of encoder choice in the overall effectiveness of the system.

Table 4: Ablation studies on retrieval performance of BrowseNet.

|  | Alternatives | HotpotQA | | 2WikiMQA | | MuSiQue | | Average | |
|---|---|---|---|---|---|---|---|---|---|
|  |  | R@2 | R@5 | R@2 | R@5 | R@2 | R@5 | R@2 | R@5 |
| BrowseNet |  | 83.85 | 96.40 | 76.77 | 93.30 | 55.12 | 73.91 | 71.91 | 87.87 |
| Synonymity threshold | 0.8 | 83.60 | 96.30 | 76.77 | 93.00 | 55.01 | 73.82 | 71.79 | 87.71 |
|  | 0.7 | 83.60 | 96.20 | 76.77 | 92.97 | 55.19 | 73.99 | 71.85 | 87.72 |
| Keyword generation | Claude-3.7-Sonnet | 83.80 | 96.40 | 76.77 | 93.10 | 55.16 | 74.21 | 71.91 | 87.80 |
|  | GPT-4o | 83.45 | 96.05 | 76.50 | 92.75 | 55.21 | 74.04 | 71.72 | 87.61 |
| n_subgraphs | 10 | 83.65 | 96.20 | 76.77 | 93.00 | 55.26 | 73.92 | 71.89 | 87.71 |
|  | 15 | 83.55 | 96.00 | 76.77 | 92.95 | 55.24 | 73.94 | 71.85 | 87.63 |
| Subquery decomposer | DeepSeek Reasoner | 83.85 | 95.80 | 76.60 | 93.05 | 56.04 | 74.33 | 72.16 | 87.73 |
|  | GPT-4o-mini | 82.85 | 95.40 | 76.05 | 92.60 | 56.77 | 74.41 | 71.89 | 87.47 |
|  | Claude-3.7-Sonnet | 80.90 | 94.05 | 75.85 | 92.12 | 53.86 | 72.95 | 70.20 | 86.37 |
| Encoder model | GTE-Qwen2 (7B) | 75.40 | 89.80 | 73.25 | 91.10 | 48.69 | 64.81 | 65.78 | 81.90 |
|  | Granite-125M-Eng. | 73.95 | 88.85 | 73.47 | 89.95 | 49.69 | 65.38 | 65.70 | 81.39 |

## 5.5 ROBUST RETRIEVAL GAINS ACROSS ENCODERS

To check if the performance gains of BrowseNet over the NaiveRAG are robust across all the encoders, we performed the robustness analysis on recall improvements. Table 5 illustrates that BrowseNet consistently outperforms NaiveRAG across all tested encoder models in terms of retrieval performance. This consistent improvement underscores that the strength of BrowseNet lies not in reliance on a particular encoder, but in the robustness of its retrieval methodology. The substantial gains observed up to 9.63 points in average Recall@5 demonstrate the effectiveness of BrowseNet's structured query decomposition and graph-based context selection, regardless of the underlying encoder used.

## 5.6 ERROR ANALYSIS

The downstream performance of our question-answering pipeline, specifically answer generation, relies on the effectiveness of the entire workflow. To analyze the source of errors, we sampled 100 questions where BrowseNet failed from the MuSiQue dataset and classified the errors into four categories: *graph-of-chunks construction*, *query-subgraph extraction*, *retrieval*, and *answer generation*. Table 6 presents the number of questions corresponding to each category. Note that a single question can be associated with multiple sources of error. The analysis reveals that nearly

Table 5: Retrieval performance comparison of NaiveRAG and BrowseNet with different encoders using Recall@5. The last column shows the gain in average Recall@5 for BrowseNet over NaiveRAG.

| Method | Encoder | HotpotQA | 2WikiMQA | MuSiQue | Average | Gain |
|--------|---------|----------|----------|---------|---------|------|
| NaiveRAG | NV-Embed-v2(7B) | 95.65 | 76.72 | 69.85 | 80.74 | – |
| BrowseNet | NV-Embed-v2(7B) | 96.40 | 93.30 | 73.91 | 87.87 | **+7.13** |
| NaiveRAG | GTE-Qwen2(7B) | 87.50 | 73.25 | 63.45 | 74.73 | – |
| BrowseNet | GTE-Qwen2(7B) | 89.80 | 91.10 | 64.81 | 81.90 | **+7.17** |
| NaiveRAG | Granite-125M-English | 85.30 | 70.62 | 59.37 | 71.76 | – |
| BrowseNet | Granite-125M-English | 88.85 | 89.95 | 65.38 | 81.39 | **+9.63** |

half of the errors originate from the semantic retrieval stage, followed by query decomposition, with smaller contributions from Graph-of-chunks construction and final answer generation. Detailed examples and case studies are provided in the Appendices A.9–A.10.

Table 6: Error source for 100 sampled questions from the MuSiQue dataset where BrowseNet produced an F1 score of zero. ME: Manual Evaluation, II: Isomorphic inaccuracy

| Stage | Graph construction | Query-subgraph extraction (ME) | Query-subgraph extraction (II) | Semantic retrieval | Answer generation |
|-------|--------------------|-------------------------------|-------------------------------|--------------------|-------------------|
| **Error %** | 9 | 33 | 35 | 49 | 9 |

## 6 CONCLUSIONS

We presented BrowseNet, a graph–based associative memory framework for multi-hop question answering. Unlike standard RAG pipelines that retrieve isolated text chunks, BrowseNet formulates retrieval as query-specific subgraph exploration. By combining lexical relationships with semantic embeddings, it constructs reasoning pathways that mirror the structural and semantic requirements of complex queries. Our experiments demonstrate that BrowseNet outperforms both dense retrieval and graph-enhanced RAG baselines, achieving state-of-the-art results on multi-hop question answering tasks. The framework reduces reliance on repeated LLM interactions by leveraging pre-generated sub-queries, making retrieval both more efficient and cost-effective. Future work will focus on expanding the range of semantic relationships encoded in the graph and extending the approach to more diverse and heterogeneous information sources, further strengthening BrowseNet as a scalable associative memory system for LLMs.

## REPRODUCIBILITY STATEMENT

All code and datasets required to reproduce our results are included in the supplementary material. A detailed README.md file is provided, outlining step-by-step instructions to replicate every experiment and result presented in the paper. We have made every effort to ensure reproducibility by independently verifying that all reported results can be faithfully reproduced using the provided resources. The benchmark dataset used in our analysis is also included in the supplementary material. Since the evaluation involves randomly sampled questions from the validation split, we have uploaded the entire corpus along with the corresponding questions to enable exact replication. Details of the computational environment and resource requirements are provided in Appendix A.12.

## ACKNOWLEDGMENTS

The authors thank Dr. Alka Bhushan for critically reviewing and commenting on the manuscript. This project is supported by funding provided by DevRev, India.

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

# A    APPENDIX

## A.1    LLM PROMPTS USED IN THE STUDY

The LLM prompt used for NER using GPT-like models is shown in Fig. 4. Prompts used for query-subgraph extraction are shown in Figs. 5, 6, 7. Prompt used for answer generation after the context retrieval is shown in Fig. 8.

## A.2    GRAPH-OF-CHUNKS EVALUATION BASED ON PRESENCE OF EDGES

For each multi-hop question in the MuSiQue dataset and 2WikiMQA, the evidence to track the reasoning path across chunks is investigated and the exact approach to track the reasoning path is discussed in this section.

### A.2.1    MUSIQUE DATASET:

In the MuSiQue dataset, for each of the questions, question decomposition is provided, and hence, the reasoning path can be traced to get the edges between the chunks. For example, for the question, "When was the person who Messi's goals in Copa del Rey compared to getting signed by Barcelona?" The question decomposition is provided as

> Q1  To whom was Messi's goal in the first leg of the Copa del Rey compared? [Chunk ID: 1]
>
> Q2  When was <ANS-1> signed by Barcelona? [Chunk ID: 2]

Using the details of Chunk IDs for the respective chunks, it can be inferred that Chunk IDs 1 and 2 should have an edge between them. In a similar fashion ground truth chunk edges (True edges) are inferred for each of the questions in the MuSiQue dataset. The edge accuracy of a question is calculated as follows

$$Edge\ accuracy = \frac{|\ \text{True edges} \cap \text{Edges in the graph-of-chunks}\ |}{|\ \text{True edges}\ |} \qquad (3)$$

For the column 'Edge Accuracy' shown in Table 1, the average of all the edge accuracies across the questions is shown.

### A.2.2    2WIKIMQA DATASET:

In the 2WikiMQA dataset, all questions are categorized into four classes: comparison, inference, composition, and bridge comparison. Comparison, inference, and composition questions involve two-hop reasoning. Inference and composition questions require a connecting edge between them, whereas comparison questions do not establish any edge between the two retrieved chunks. In contrast to them, bridge comparison questions involve four-hop reasoning with two connecting edges. Given the predefined question types, the true edges in this dataset can be inferred. The edge accuracy is computed using the formula provided in Equation 3.

## A.3    ISOMORPHIC ACCURACY

Isomorphic accuracy calculated for the query-subgraphs is discussed in this section. Consider the following multi-hop query from the MuSiQue dataset:

> *"What month did the Tripartite discussions begin between Britain, France, and the country where, despite being headquartered in the nation called the nobilities commonwealth, the top-ranking Warsaw Pact operatives originated?"*

The gold query decomposition is:

- **Q1:** What was the nobilities commonwealth?
- **Q2:** Despite being headquartered in #1, the top-ranking operatives of the Warsaw Pact were from which country?

- **Q3:** What month did the Tripartite discussions begin between Britain, #2, and France?

This decomposition can be represented as a graph: **Q1** → **Q2** → **Q3**.

In contrast, the query decomposition generated by GPT-4o is:

- **Q1:** What is the nation called the nobility's commonwealth?
- **Q2:** <Q1> Where are the top-ranking Warsaw Pact operatives headquartered?
- **Q3:** <Q2> In which country did the top-ranking Warsaw Pact operatives originate?
- **Q4:** <Q3> What month did the Tripartite discussions begin between Britain, France, and the country where the top-ranking Warsaw Pact operatives originated?

This decomposition forms a graph: **Q1** → **Q2** → **Q3** → **Q4**.

Since the generated and gold decompositions differ in structure, they are not isomorphic, resulting in an isomorphic accuracy score of **0**.

A similar evaluation procedure is applied to the 2WikiMQA dataset. The average isomorphic accuracy across models is summarized in Table 7.

Table 7: Isomorphic accuracy of generated query decompositions.

| LLM Model | 2WikiMQA | MuSiQue |
|---|---|---|
| GPT-4o | 0.973 | 0.685 |
| Claude-3.7-Sonnet | 0.967 | 0.487 |

The results indicate that isomorphic accuracy is consistently higher on the 2WikiMQA dataset than on MuSiQue for both models. GPT-4o performs slightly better than Claude-3.7 Sonnet across both datasets, achieving 0.973 on 2WikiMQA and 0.685 on MuSiQue, compared to 0.967 and 0.487, respectively.

### A.4 GRAPH DENSITY CALCULATION

The density of an undirected graph is a measure of how many edges are in the graph compared to the maximum possible number of edges. Let $G = (V, E)$ be a graph. The density $GD$ of the graph is given by:

$$GD = \frac{2|E|}{|V|(|V| - 1)}$$

where $|\cdot|$ indicates the cardinality of the given set.

### A.5 IMPORTANCE OF GRAPH-OF-CHUNKS

The graph-of-chunks plays a vital role in BrowseNet, not only in capturing dependencies but also in enhancing latency. In BrowseNet, it is used for the dynamic modification of the corpus, which constitutes the foundation of the proposed approach. This dynamic refinement significantly improves both retrieval effectiveness and latency. Table 8 shows the improvement in retrieval performance when the graph-of-chunks is utilized, compared to a baseline that considers all nodes as the corpus at every retrieval step using the subquery. The results in the table are reported on a sample of 100 questions from the MuSiQue dataset.

Table 8: Impact of using Graph-of-chunks on retrieval performance and latency for 100 questions from the MuSiQue dataset.

| Setting | Time Taken (sec) | Recall@5 |
|---|---|---|
| Without graph-of-chunks | 177 | 70.75 |
| With graph-of-chunks | 115 | 75.08 |

## A.6 LATENCY ANALYSIS

To evaluate latency, we sampled 50 questions from the MuSiQue dataset. Table 9 presents a comparison between BrowseNet and HippoRAG-2 based on the Average Time Per Query (ATPQ) metric, which measures the average time in seconds taken to process each query up to the retrieval stage. As shown in Table 9, BrowseNet incurs a slightly higher latency compared to HippoRAG-2. However, the additional latency is marginal (0.49 seconds on average) and is justified by the significant gains in retrieval accuracy and overall QA performance, as demonstrated in previous sections.

Table 9: Latency comparison between BrowseNet and HippoRAG-2 based on the Average Time Per Query (ATPQ) for 50 sampled questions from the MuSiQue dataset.

| Method | ATPQ (seconds) |
|---|---|
| BrowseNet | 2.70 |
| HippoRAG-2 | 2.21 |

For BrowseNet, query decomposition accounts for 1.50 seconds of the total latency, primarily due to the overhead of API-based LLM calls. This component could be significantly improved by deploying the language model locally, which would reduce network latency. In contrast, the retrieval stage is relatively efficient, taking only 1.19 seconds on average.

## A.7 COST ANALYSIS

We provide a quantitative comparison of LLM-related costs between BrowseNet and the state-of-the-art retrieval baseline, HippoRAG-2, using the HotpotQA benchmark dataset in Table 10. The analysis considers the full pipeline cost, from indexing to retrieval, using `gpt-4o-mini` in both systems. The pricing model follows OpenAI's current API rates (As of $24^{th}$ September 2025): \$0.15 per 1M input tokens and \$0.6 per 1M output tokens.

Table 10: Token-level comparison between HippoRAG-2 and BrowseNet on HotpotQA.

| | HippoRAG-2 | BrowseNet |
|---|---|---|
| Input tokens | 5,880,618 | 249,503 |
| Output tokens | 2,110,007 | 44,641 |
| Total tokens | 7,990,625 | 294,144 |

The corresponding LLM costs can be estimated as follows:

$$\text{Cost} = \frac{\text{Input tokens}}{10^6} \times 0.15 + \frac{\text{Output tokens}}{10^6} \times 0.6$$

Using this formula:

- HippoRAG-2: $\approx$ \$2.15
- BrowseNet: $\approx$ \$0.064

Thus, BrowseNet achieves roughly 33$\times$ higher cost efficiency while maintaining state-of-the-art retrieval.

## A.8 ABLATIONS

We performed ablation studies on the graph-of-chunks construction by varying the NER model for keyword extraction and the ColBERT synonymity threshold for connecting nodes. As shown in Table 11, edge accuracy on the 2WikiMQA dataset remains near 100% across all thresholds,

indicating robustness to synonymity settings. In contrast, for the MuSiQue dataset, higher edge accuracy is observed at lower synonymity thresholds across all NER models.

We also conducted ablations on answer generation using different LLMs (Table 12). The results indicate a substantial variation in performance, with average exact match scores differing by approximately 10% between models, highlighting the influence of the LLM choice on final retrieval quality.

Table 11: Graph-of-chunks performance with varying ColBERT synonymity threshold. Here, GD refers to Graph density, and EA refers to Edge accuracy.

| NER Model | Synonymity Threshold | HotpotQA | | 2WikiMQA | | MuSiQue | |
|---|---|---|---|---|---|---|---|
| | | GD | EA | GD | EA | GD | EA |
| GLiNER | 0.9 | 0.0641 | NA | 0.0978 | 99.86 | 0.0498 | 91.03 |
| | 0.8 | 0.2309 | NA | 0.2653 | 99.86 | 0.1732 | 95.18 |
| | 0.7 | 0.3781 | NA | 0.4018 | 100 | 0.3155 | 97.43 |
| GPT-4o | 0.9 | 0.0844 | NA | 0.0663 | 98.74 | 0.0371 | 97.83 |
| | 0.8 | 0.2533 | NA | 0.2163 | 98.94 | 0.1652 | 97.94 |
| | 0.7 | 0.3976 | NA | 0.3282 | 99.20 | 0.2934 | 98.71 |
| Claude-3.7-Sonnet | 0.9 | 0.0893 | NA | 0.0673 | 99.74 | 0.0550 | 94.33 |
| | 0.8 | 0.2667 | NA | 0.2325 | 99.87 | 0.1848 | 98.26 |
| | 0.7 | 0.4086 | NA | 0.3399 | 100 | 0.3143 | 98.54 |

Table 12: Performance comparison of different LLMs for answer generation on HotpotQA, 2WikiMQA, and MuSiQue. The best score is written in **Bold** and the second best is underlined.

| LLM | HotpotQA | | 2WikiMQA | | MuSiQue | | Average | |
|---|---|---|---|---|---|---|---|---|
| | EM | F1 | EM | F1 | EM | F1 | EM | F1 |
| gpt-4o-mini | 62.20 | 77.69 | 63.90 | 74.50 | 41.60 | 54.08 | 55.90 | 68.76 |
| gpt-3.5-turbo | 58.80 | 73.81 | 47.70 | 59.57 | 37.40 | 49.77 | 47.97 | 61.05 |
| gpt-4.1-mini | 63.20 | **79.21** | 64.50 | 74.43 | 42.70 | 55.07 | 56.80 | 69.57 |
| deepseek-chat-v3 | 62.20 | 78.91 | **66.10** | **75.86** | **43.50** | **56.25** | **57.27** | **70.34** |
| gemini-2.0-flash | **63.40** | 78.00 | 62.10 | 70.30 | 38.10 | 47.37 | 54.53 | 65.22 |

## A.9 Case Study

Using example questions, we demonstrate how BrowseNet improves retrieval performance compared to the NaiveRAG approach. Table 13 presents a comparison of the articles retrieved by each method. As we can see for NaiveRAG, the retrieved corpus is related to the keywords present in the query rather than the underlying reasoning path required to answer the multi-hop question, which often leads to the inclusion of spurious or contextually irrelevant passages.

For a detailed understanding of the BrowseNet approach, consider the question, 'What is the Till dom ensamma performer's birth date?'. The query decomposition produces the subqueries:

- Q1) Who is the performer of "Till dom ensamma"?
- Q2) <Q1> What is the birth date of the performer of "Till dom ensamma"?

The subgraph to be retrieved from the graph-of-chunks has the structure, **Q1 → Q2**. Retrieval begins with the initiator node Q1, treating all nodes in the graph-of-chunks as the initial corpus. Using semantic similarity search, the top 5 most similar nodes (passages) are retrieved. Among them, the passage containing "Till dom ensamm" is found to be most similar to Q1.

Next, for the query node Q2, the candidate contexts is restricted to the neighbors of the previously retrieved passage "Till dom ensamma". Then Q2 is used to search the new corpus, from which the passage "Mauro Scocco" is identified as most relevant and subsequently retrieved.

Table 13: Top-5 articles retrieved by NaiveRAG and BrowseNet for each question. The underlined entries indicate the golden articles required to answer the corresponding question.

| Question | Top-5 Retrieved Documents | |
|---|---|---|
| | **NaiveRAG** | **BrowseNet** |
| What is the Till dom ensamma performer's birth date? | 1. Till dom ensamma
2. Langa Natter
3. Nar hela varlden ser pa
4. Roxy Recordings
5. Du far gora som du vill | 1. Till dom ensamma
2. Mauro Scocco
3. Hits (Mauro Scocco album)
4. Langa natter
5. Nar hela varlden ser pa |
| How many episodes are in season 5 of the series with "The Bag or the Bat"? | 1. The Bag or the Bat
2. List of Orange Is the New Black episodes
3. Cheatty Cheatty Bang Bang
4. Samurai Jack (season 5)
5. Arrested Development (season 5) | 1. The Bag or the Bat
2. List of Ray Donovan episodes
3. List of Orange Is the New Black episodes
4. Cheatty Cheatty Bang Bang
5. Samurai Jack (season 5) |

Table 14: Number of entities extracted using distinct NER models.

| NER Model | No. of Entities | | |
|---|---|---|---|
| | HotPotQA | 2WikiMQA | MuSiQue |
| GLiNER | 60,862 | 44,907 | 67,332 |
| GPT-4o | 61,959 | 41,219 | 66,795 |
| Claude-3.7 | 62,671 | 41,921 | 67,216 |

## A.10 ERROR ANALYSIS: DETAILS

- **Graph-of-chunks construction:** These errors arise because of missing edges in the constructed graph. Our analysis reveals that 9% of relevant edges are missing from the graph.

- **Query-subgraph extraction:** This stage is evaluated through two approaches: manual analysis to determine whether each question is correctly decomposed, and isomorphic accuracy, as defined in Section 4.3. Human evaluation shows that 33% of the questions are either incorrectly decomposed or contain redundant sub-queries (i.e., multiple sub-queries retrieving the same information). Isomorphic accuracy reveals that 35% of the questions result in query-subgraphs that are not isomorphic to the ground-truth structure, indicating errors or redundancies in the decomposition process.

  Here, we present an example where query-graph generation failed to produce accurate outputs.

  **Query:** Are both businesses, Google and Banco De Ponce, located in the same country? The generated single-hop queries with dependencies are:

  - **Q1:** In which country is Google located?
  - **Q2:** In which country is Banco De Ponce located?
  - **Q3:** <ANS-1> <ANS-2> Are both Google and Banco De Ponce located in the same country?

  In this case, only the first two questions are necessary to traverse the graph. The third question is redundant because the answer to the third question cannot be inferred from any of the chunks in the graph-of-chunks.

- **Semantic retrieval:** In terms of overall recall (Recall@5) distribution across all evaluated questions, 1% of the questions resulted in zero recall, while 48% exhibited a recall of less than 1. To understand the influence of query-subgraph extraction on retrieval effectiveness, we analyze recall performance based on both human evaluation and isomorphic accuracy.

According to human evaluation, among the 33 incorrectly decomposed questions, only 11 (33.33%) achieved full recall. In contrast, for the 67 correctly decomposed questions, 40 (59.70%) achieved full recall. This highlights the critical role of accurate query decomposition in successful retrieval. A similar trend is observed using isomorphic accuracy. Out of the 35 incorrectly decomposed questions (based on non-isomorphic subgraphs), 12 (34.28%) achieved full recall. Among the 65 correctly decomposed questions, 40 (60.00%) achieved full recall. These results further reinforce that effective query decomposition significantly improves retrieval performance.

- **Answer Generation:** For questions where the retrieval stage achieved full recall, the final answer generation step still produced incorrect answers in 9% of the cases. This indicates that even when all relevant information is successfully retrieved, the model may still fail to generate the correct answer. Such failures can be attributed to challenges in reasoning, interpreting the evidence, or inherent limitations of the generative model itself.

Importantly, despite errors in decomposition, BrowseNet's adaptive retrieval strategy was still able to achieve full recall in approximately one-third of the incorrectly decomposed cases. This demonstrates the robustness of the system in handling imperfect inputs. The utilization of a multi-query approach, as discussed in Section 3.2.2, plays a key role in alleviating the impact of decomposition errors and enhancing retrieval robustness.

## A.11 STATISTICAL SIGNIFICANCE

Confidence intervals for retrieval results are presented in Table 15, computed using a paired bootstrap test with 10,000 resamples and $\alpha = 0.05$, yielding 95% confidence intervals. We also performed significance analysis across all retrieval comparisons. Most of the BrowseNet retrieval results are statistically significant ($p < 0.05$), except in three cases: HotpotQA-Recall@2 vs. NV-Embed-v2, HotpotQA-Recall@5 vs. HippoRAG-2, and MuSiQue-Recall@5 vs. HippoRAG-2, where the improvements are not statistically significant. However, even when retrieval differences are small, BrowseNet still achieves stronger overall question-answering performance because the query-subgraph guides the LLM more effectively toward the correct final answer.

Table 15: Confidence Intervals (CIs) of recall metric for context retrieval on HotpotQA, 2WikiMQA, and MuSiQue. The best score is written in **Bold** and the second best is underlined. The confidence intervals are represented in the superscript (upper bound) and subscript (lower bound) of the mean value.

| Retriever | HotpotQA | | 2WikiMQA | | MuSiQue | |
|---|---|---|---|---|---|---|
| | R@2 | R@5 | R@2 | R@5 | R@2 | R@5 |
| NV-Embed-v2 (7B) | $\mathbf{83.95}^{+1.60}_{-1.60}$ | $95.64^{+0.86}_{-0.89}$ | $69.05^{+1.57}_{-1.53}$ | $76.72^{+1.51}_{-1.47}$ | $53.31^{+1.67}_{-1.65}$ | $69.86^{+1.70}_{-1.71}$ |
| HippoRAG-2 | $81.79^{+1.61}_{-1.69}$ | $\underline{96.20}^{+0.85}_{-0.90}$ | $\underline{75.45}^{+1.62}_{-1.68}$ | $\underline{90.62}^{+1.11}_{-1.09}$ | $\underline{53.63}^{+1.82}_{-1.81}$ | $\underline{73.70}^{+1.69}_{-1.68}$ |
| BrowseNet | $\underline{83.83}^{+1.52}_{-1.58}$ | $\mathbf{96.40}^{+0.80}_{-0.85}$ | $\mathbf{76.77}^{+1.66}_{-1.64}$ | $\mathbf{93.29}^{+0.91}_{-0.91}$ | $\mathbf{55.12}^{+1.76}_{-1.68}$ | $\mathbf{73.92}^{+1.69}_{-1.67}$ |

## A.12 COMPUTATIONAL RESOURCES USED:

To get the embeddings from the encoder, NV-Embed-v2, we have used NVIDIA A100 GPU and 512GB RAM.

## A.13 LARGE LANGUAGE MODEL (LLM) USAGE:

During the preparation of this work, the author(s) used Large Language Model application to improve the paper's organizational flow and eliminate errors by providing the draft. After using this tool or service, the author(s) reviewed and edited the content as needed and take(s) full responsibility for the content of the published article.

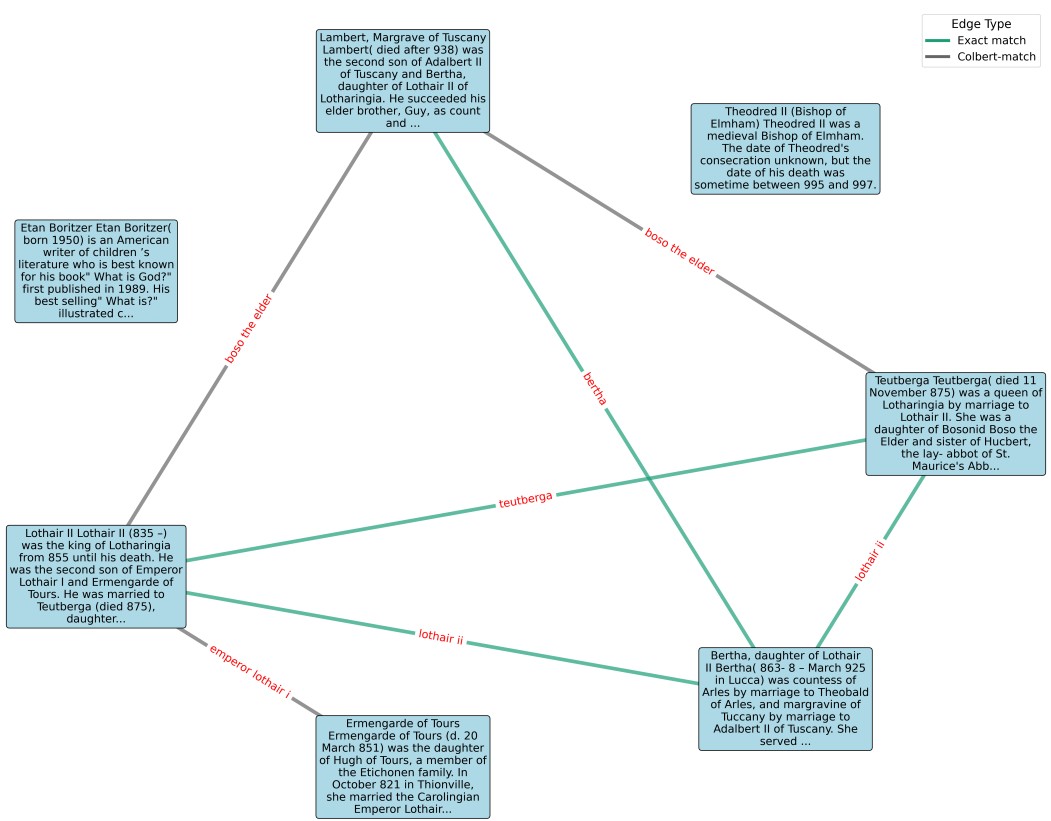

Figure 2: A small snippet (seven nodes) of graph-of-chunks obtained from the 2WikiMultiHopQA. The color of the edge denotes whether the keyword obtained is from an exact match or derived from the Colbert similarity mapping.

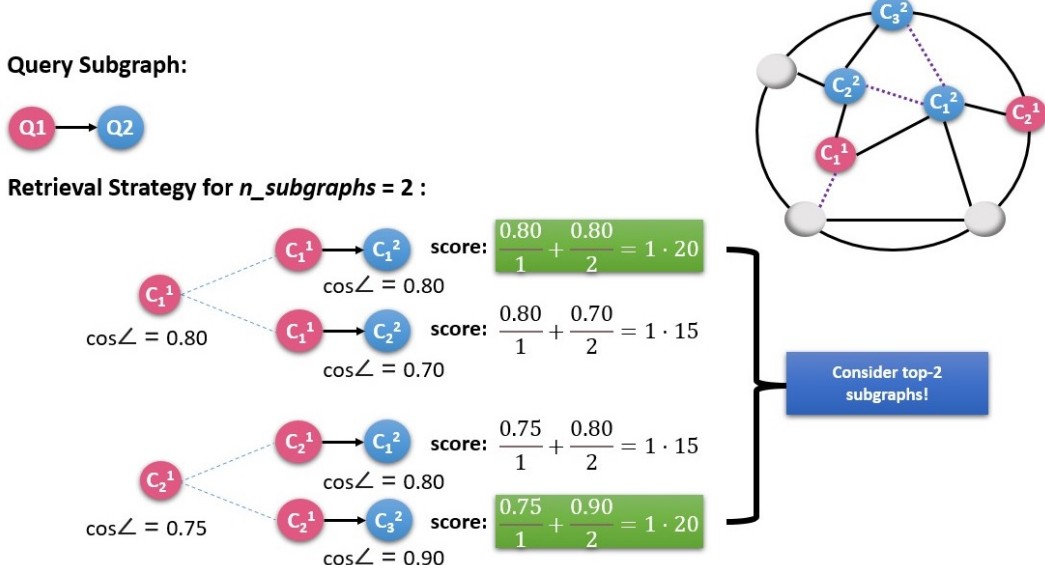

Figure 3: Traversal over the graph-of-chunks is analogous to the beam search where the top-K candidates are retained at every iteration. Here $cos\angle$ refers to the maximum of cosine similarity obtained from Equation 1, $c_i^k$ refers to the candidate chunks to be retrieved from the graph-of-chunks.

---

**Prompt:**

Your task is to extract named entities from the given paragraph.    Respond with a JSON list of entities.

**One-shot demonstration:**
**Input:**
Radio City is India's first private FM radio station and was started on 3 July 2001.    It plays Hindi, English and regional songs.    Radio City recently forayed into New Media in May 2008 with the launch of a music portal - PlanetRadiocity.com that offers music related news, videos, songs, and other music-related features.

**Output:**
{"named_entities":      ["Radio City", "India", "3 July 2001", "Hindi", "English", "May 2008", "PlanetRadiocity.com"]}

---

Figure 4: LLM prompt used for named entities extraction. This prompt is adapted from the HippoRAG (Gutiérrez et al., 2024)

---

**Prompt:**

You are a helpful assistant designed to split a multi-hop query into a set of single-hop queries. These smaller queries will be used later to retrieve relevant information from a corpus in an automated manner.Each query (Q1, Q2, etc.) should progressively build upon the answer to the previous ones except for the comparison kind of questions. The follow-up questions should have an indicator of the previous question they are building upon (like <Q1>). Few-shot examples are provided below for reference and follow the similar format.

**Few-shot demonstration:**

**INPUT**: 'Who married the publisher of abolitionist newspaper The North Star?'
**OUTPUT**: Q1) Who is the publisher of abolitionist newspaper The North Star? Q2) <Q1> Who married the publisher of abolitionist newspaper The North Star?

**INPUT**: 'In what state is the district where the man who wanted to reform the religion practiced by Innocenzo Ferrieri preached a sermon on Marian devotion?'
**OUPUT**: Q1) What is the religion of Innocenzo Ferrieri? Q2) <Q1> Who wanted to reform the religion practiced by Innocenzo Ferrieri? Q3) <Q2> What is the district where the man who wanted to reform the religion practiced by Innocenzo Ferrieri preached a sermon on Marian devotion?Q4) <Q3> In what state is the district where the man who wanted to reform the religion practiced by Innocenzo Ferrieri preached a sermon on Marian devotion?

**INPUT**: 'The Beach was filmed in what location of the country that contains the birth city of Siddhi Savetsila?'
**OUTPUT**: Q1) What is the birth city of Siddhi Savetsila? Q2) <Q1> In which country is the birth city of Siddhi Savetsila located? Q3) <Q2> The Beach was filmed in what location of the country that contains the birth city of Siddhi Savetsila?"'

---

Figure 5: LLM prompt used for query-subgraph extraction in the Musique dataset. Few shot example questions are taken from Musique dataset

---

**Prompt:**

You are a helpful assistant designed to split a multi-hop query into a set of single-hop queries. These smaller queries will be used later to retrieve relevant information from a corpus in an automated manner.Each query (Q1, Q2, etc.) should progressively build upon the answer to the previous ones except for the comparison kind of questions. The follow-up questions should have an indicator of the previous question they are building upon (like <Q1>). Few-shot examples are provided below for reference and follow the similar format.

**Few-shot demonstration:**

**INPUT**: 'Which film was released first, Aas Ka Panchhi or Phoolwari?'
**OUTPUT**: Q1) When was the film Aas Ka Panchhi released?
Q2) When was the film Phoolwari released?

**INPUT**: 'Which film has the director who died first, The Goose Woman or You Can No Longer Remain Silent?'
**OUTPUT**: Q1) Who is the director of The film Goose Woman?
Q2) Who is the director of the film You Can No Longer Remain Silent?
Q3) <Q1> When did the director of The film Goose Woman die?
Q4) <Q2> When did the director of the film You Can No Longer Remain Silent die?

**INPUT**: 'Who lived longer, Ludwig Elsbett or Pamela Ann Rymer?'
**OUTPUT**: Q1) How long did Ludwig Elsbett live?
Q2) How long did Pamela Ann Rymer live?

**INPUT**: 'What is the place of birth of the director of film Gaby: A True Story?'
**OUTPUT**: Q1) Who is the director of film Gaby: A True Story?
Q2) <Q1> What is the place of birth of the director of film Gaby: A True Story?

**INPUT**: 'Who is the father-in-law of Sisowath Kossamak?'
**OUTPUT**: Q1) Who husband/wife of Sisowath Kossamak?
Q2) <Q1> Who is the father-in-law of Sisowath Kossamak?

---

Figure 6: LLM prompt used for query-subgraph extraction in the 2WikiMQA dataset. Few shot example questions are taken from 2WikiMQA dataset

**Prompt:**

You are a helpful assistant designed to split a multi-hop query into a set of single-hop queries. These smaller queries will be used later to retrieve relevant information from a corpus in an automated manner. Each query (Q1, Q2, etc.) should progressively build upon the answer to the previous ones except for the comparison kind of questions. The follow-up questions should have an indicator of the previous question they are building upon (like <Q1>). Few-shot examples are provided below for reference and follow the similar format.

**Few-shot demonstration:**

**INPUT**: 'What was the other name for the war between the Cherokee people and white settlers in 1793?'
**OUTPUT**: Q1) What was the war between the Cherokee people and white settlers in 1793 called? Q2) <Q1> What was the other name for the war between the Cherokee people and white settlers in 1793?

**INPUT**: 'Which university has more campuses, Dalhousie University or California State Polytechnic University, Pomona?'
**OUTPUT**: Q1) How many campuses does Dalhousie University have? Q2) How many campuses does California State Polytechnic University, Pomona have?

Figure 7: LLM prompt used for query-subgraph extraction in the HotpotQA dataset. Few shot example questions are taken from HotpotQA dataset

**Prompt:**

As an advanced reading comprehension assistant, your task is to analyze the retrieved context wikipedia passages to answer the Question meticulously. To arrive at the final answer, use the subqueries provided to you in the given order. If you cannot answer a subquery, then try to answer the question to the best of your ability, based on the information available in the retrieved context. Start your response with "Thought: " where you systematically explain your reasoning process, breaking down the steps leading to the answer. Conclude with "Answer: " followed by a concise, definitive response limited to just the essential words, no extra sentences or explanations. In case of multiple possible answers, provide answers with (or) in between, such as "American (or) America (or) United States of America".

NOTE:
1. I hope your answer matches the answer exactly, so ENSURE that the answer following "Answer:" is concise, such as 14 May, 1832 or yes. THE SHORTER, THE BETTER!
2. If the answer is a date, please provide the full date as much as possible, such as 18 May, 1932.
3. If the answer is a place, please provide the full name of the place as much as possible, such as "Oxford, England" instead of "Oxford".

Figure 8: LLM prompt used for answer generation after relevant context retrieval

**Algorithm 1** Graph Traversal and Context Retrieval for Query-Subgraph

---

1: **Input:** Original Query $Q_{orig}$, Query-subgraph $\boldsymbol{Q} = (\boldsymbol{V_q}, \boldsymbol{E_q})$ with connected components $\{Q^{(1)}, Q^{(2)}, \dots\}$, Graph-of-chunks $\boldsymbol{G} = (\boldsymbol{V}, \boldsymbol{E})$, number of subgraphs $k$, Encoder $\boldsymbol{M}$

2: **Output:** Top-$k$ scored subgraphs per connected component as retrieved context

3: **for** each connected component $Q^{(i)}$ $in$ $\boldsymbol{Q}$ **do**

4:     $\{V_q^1, V_q^2, \dots\} \leftarrow Topological\_sort(Q^{(i)})$

5:     $weight(c) = 0$ for each $c \in V$

6:     **for** each subquery $V_q^i \in \{V_q^1, V_q^2, \dots\}$ **do**

7:         **if** $V_q^i$ has no predecessors **then**

8:             Retrieve top-$k$ chunks $c_1, c_2, \dots, c_k \in \boldsymbol{V}$ based on Equation 1

9:             add top-$k$ selected chunks $c_1, c_2, \dots, c_k$ in $CandidateSet(V_q^i)$

10:             **for** each $c \in CandidateSet(V_q^i)$ **do**

11:                 $weight(c) = SS_c$

12:             **end for**

13:         **else**

14:             Let $P = \{V_q^j : V_q^j \text{ is a predecessor of } V_q^i\}$

15:             Generate all combinations $\mathcal{C}$ by selecting one candidate from each predecessor in $P$

16:             **for** each combination $\mathcal{C}_n \in \mathcal{C}$ **do**

17:                 $Neighbours(C_n) = \bigcup_{c_j \in \mathcal{C}_n} Neighbours(c_j) \cup (\mathcal{C} \setminus \mathcal{C}_n)$

18:                 Set $V_q^{i'} = V_q^i$

19:                 **for** each $c_j \in \mathcal{C}_n$ **do**

20:                     **if** $\cos\angle(V_q^i, c_j) > \max(\{\cos\angle(V_q^i, cnk) : cnk \in \bigcup_{c_j \in \mathcal{C}_n} Neighbours(c_j)\})$ **then**

21:                       $V_q^{i'} \leftarrow concat(c_j, V_q^{i'})$

22:                     **end if**

23:                 **end for**

24:                 **for** each $v \in Neighbors(C_n)$ **do**

25:                     Compute $SS_v = \max(\cos\angle(V_q^i, v), \cos\angle(V_q^{i'}, v), \cos\angle(Q_{orig}, v))$

26:                 **end for**

27:                 Select top-$k$ chunks $K$ from $Neighbours(C_n)$ based on $SS_v$

28:                 **for** each $c_k \in K$ **do**

29:                     $wt(c_k) = \frac{SS_{c_k}}{depth(V_q^i)}$

30:                     **for** each $c_n \in \mathcal{C}_n$ **do**

31:                       update weight of $c_k$ $wt(c_k) = wt(c_k) + weight(c_n)$

32:                     **end for**

33:                     **if** $weight(c_k) < wt(c_k)$ **then**

34:                       $weight(c_k) = wt(c_k)$

35:                       update predecessors of $c_k$ as $C_n$ and their predecessor list

36:                       add $c_k$ in $CandidateSet(C)$ with $weight_{c_k}$

37:                   **end if**

38:                 **end for**

39:                 Select top-$k$ chunks from $CandidateSet(C)$ and include in $CandidateSet(V_q^i)$

40:                 **if** $V_q^i$ is terminal node **then**

41:                   Selected chunks at the terminal nodes along with their predecessors are the selected top-$k$ subgraphs for $Q^{(i)}$

42:                 **end if**

43:             **end for**

44:         **end if**

45:     **end for**

46: **end for**

---

