# OpenReview forum: "BrowseNet: Graph-Based Associative Memory for Contextual Information Retrieval"
_ICLR.cc/2026/Conference — ICLR 2026 Poster_

### Official Review · Reviewer_Dpwg · 2025-10-22

**Soundness:** 2
**Presentation:** 2
**Contribution:** 2
**Rating:** 2
**Confidence:** 4

**Summary:**

This paper presents an implementation of GraphRAG. The idea is to decompose the query into a graph of sub-queries using LLM, create a graph of passages based on common enities, and use the query graph to guide the exploration of the passage graph to select chunks to be fed into LLM for generation.

**Strengths:**

S1. The paper is generally easy to read.

S2. The experiments are well designed, providing retrieval and generation performance as well as extensive ablation studies.

**Weaknesses:**

W1. Novelty is limited. KG construction (Section 3.1), query decomposition (Section 3.2.1), and answer generation (Section 3.3) are standard practice in the literature. The rest of the approach, KG traversal (Section 3.2.2), is heuristic and its generalizability is not well justified given experiments on only three datasets.

W2. Baselines are focused on recent RAG solutions, but the proposed approach is closely related to early multi-hop QA solutions, which are not compared empirically.

W3. The constructed KG is actually not a KG but a graph of passages connected based on common entities. This is a common practice in conventional multi-hop QA research. It is not a KG that should represent relations between entities.

**Questions:**

Q1. Can you compare your approach with early multi-hop QA methods that also build a graph of passages based on their similarity (e.g., common entities)?

Q2. How do you demonstrate the generalizability of your heuristic graph traversal?

---

> ### Author Response · Authors · 2025-11-22
> **Response to reviewer Dpwg**
>
> We thank the reviewer for their valuable feedback. Our response to the comments are as follows
>
> **Weakness-1:**
>
> KG construction, query decomposition, and answer generation are standard components. However, BrowseNet's innovation lies in the integration of these elements in a unified manner for (i) graph–based associative retrieval and (ii) in the specific traversal and candidate-pruning strategy used for multi-hop QA. We would like to emphasize that BrowseNet’s graph traversal is not a generic heuristic walk but a query-subgraph–guided process that combines:
> (i) LLM-based decomposition into an explicit query subgraph;
> (ii) alignment of subgraph nodes with initiator chunks via semantic similarity;
> (iii) constrained expansion over a content-centric graph rather than an entity-only KG; and
> (iv) multi-hop scoring that jointly considers local subquery relevance while ensuring consistency with the global multi-hop question.
>
> This novelty in the design of BrowseNet allows one-shot query decomposition followed by structured, bounded exploration. These steps differ from the existing KG-RAG and GraphRAG variants that typically rely on iterative LLM-in-the-loop traversal or purely similarity-based expansion. Hence, BrowseNet provides a novel framework that leverages these concepts to achieve the state-of-the-art (SOTA) performance.  Regarding generalizability, BrowseNet is evaluated on three challenging multi-hop benchmarks (HotpotQA, 2WikiMultiHopQA, MuSiQue), which together cover diverse reasoning patterns (bridge, comparison, compositional reasoning) across varied topics. Across these datasets, BrowseNet consistently outperforms strong graph- and structure-aware RAG baselines such as GraphRAG and SiReRAG in EM and F1, suggesting that the traversal strategy is not overfit to a single corpus or question style.
>
> **Weakness-2 and Quesion-1:**
>
> Our baselines focus on state-of-the-art RAG solutions that currently achieve the best performance on multi-hop QA benchmarks. The closest early multi-hop QA method is KG-Prompting [1] that has the nodes as chunks and edges are based on the keywords. This method employ a different evaluation setting: they construct one question-specific graph per query using the labeled distractors and golden passages provided for that question. In contrast, our method extends the benchmark difficulty by augmenting the candidate corpus to include all passages from other questions as distractors as mentioned in the text. This makes the retrieval task significantly harder and more realistic, where the system must select relevant context from a large, mixed set rather than a question-limited set of passages.
>
> Also, BrowseNet has been benchmarked against RAPTOR [2] in the main text. RAPTOR constructs the knowledge graph on top of the chunks and the edges reflect the similarities between them. Furthermore, we have now included a new benchmark, SiReRAG [3] to compare with BrowseNet. SiReRAG includes both kinds of graph: chunk based and entity based. The results on question-answering on comparison to SiReRAG is given in below table.
>
>
> |    Method (K=5)  | HotpotQA ||2WikiMQA||MuSiQue||Average||
> |------|------|------|------|------|------|------|------|------|
> |  | EM | F1 | EM | F1 | EM | F1 | EM | F1 |
> | Browsenet (gpt-4o-mini) | **62.20**  | **77.69**   | **63.90**   | **74.50**  | **41.60**  | **54.08**    | **55.90**   |  **68.76**  |
> | SiReRAG (gpt-4o-mini)  | 48.30  | 63.17  | 41.30   | 48.05   | 26.00   | 39.59  | 38.53  |50.27   |
>
>
> **References:**
> 1) Wang, Y., Lipka, N., Rossi, R. A., Siu, A., Zhang, R., & Derr, T. (2024, March). Knowledge graph prompting for multi-document question answering. In Proceedings of the AAAI conference on artificial intelligence (Vol. 38, No. 17, pp. 19206-19214).
> 2) Sarthi, P., Abdullah, S., Tuli, A., Khanna, S., Goldie, A., & Manning, C. D. (2024, May). Raptor: Recursive abstractive processing for tree-organized retrieval. In The Twelfth International Conference on Learning Representations.
> 3) Zhang, N., Choubey, P. K., Fabbri, A., Bernadett-Shapiro, G., Zhang, R., Mitra, P., ... & Wu, C. S. (2025). Sirerag: Indexing similar and related information for multihop reasoning. arXiv preprint arXiv:2412.06206. In The Thirteenth International Conference on Learning Representations.

---

> ### Author Response · Authors · 2025-11-22
> **Response to reviewer Dpwg (continued)**
>
> **Weakness-3**
>
> We recognize that a traditional knowledge graph involves entities as nodes and predicates as edges, while our graph is constructed over chunks of text, which suits retrieval-augmented generation tasks better. We initially followed prior work using the term "knowledge graph" [1][2], but upon reflection, we agree the term could be clearer. We will revise the manuscript to replace "knowledge graph" with "graph of chunks" wherever appropriate
>
> **Action:**
> We will revise the manuscript to replace all instances of "knowledge graph" with "graph of chunks" to improve clarity and accuracy.
>
> **References:**
> 1) Wang, Y., Lipka, N., Rossi, R. A., Siu, A., Zhang, R., & Derr, T. (2024, March). Knowledge graph prompting for multi-document question answering. In Proceedings of the AAAI conference on artificial intelligence (Vol. 38, No. 17, pp. 19206-19214).
> 2) Yang, Z., Zhu, Z., & Zhu, J. (2025, April). CuriousLLM: Elevating multi-document question answering with llm-enhanced knowledge graph reasoning. In Proceedings of the 2025 Conference of the Nations of the Americas Chapter of the Association for Computational Linguistics: Human Language Technologies (Volume 3: Industry Track) (pp. 274-286).
>
> **Question-2**
>
> The generalizability of the graph traversal in our approach is demonstrated through consistent, strong performance across three diverse and strong multi-hop QA benchmarks, HotpotQA, 2WikiMultiHopQA, and MuSiQue. These datasets encompass varied reasoning types, including bridge, comparison, and compositional questions in multiple domains. These datasets differ significantly in corpus size, question complexity, and domain, yet BrowseNet’s traversal strategy consistently outperforms competitive graph- and structure-aware retrieval baselines such as HippoRAG and SiReRAG. This empirical evidence indicates that our traversal approach effectively adapts to different reasoning requirements and corpus characteristics without being overfitted to a single dataset or domain.

---

> > ### Author Response · Authors · 2025-11-26
> > **Follow-up**
> >
> > Dear Reviewer,
> > I hope this message finds you well. As there is still about one week remaining in the discussion period, we would like to kindly follow up and ensure that our rebuttal has sufficiently addressed your concerns.
> > If there are any remaining questions or additional feedback you would like us to consider, please feel free to let us know. Your insights are highly valuable to us, and we would be glad to provide further clarification or revisions.
> >
> > Thank you again for your time and effort in reviewing our paper.
> >
> > Best regards,
> > The Authors

---

> > > ### Comment · Reviewer_Dpwg · 2025-11-27
> > >
> > > Thank you for your responses. While I am still not convinced about novelty, your response to W2 and your action to W3 are generally satisfactory. I will increase my score.

---

> > > > ### Author Response · Authors · 2025-11-28
> > > > **Response to reviewer Dpwg**
> > > >
> > > > Thank you for updating your score and for your careful consideration of our work. We appreciate your time and constructive evaluation. We would be happy to address any further comments or questions you may have and to make additional clarifications or improvements if needed.

---

### Official Review · Reviewer_9MFQ · 2025-10-28

**Soundness:** 3
**Presentation:** 3
**Contribution:** 3
**Rating:** 6
**Confidence:** 4

**Summary:**

The paper introduces BrowseNet, a framework for retrieval-augmented generation (RAG) that represents document collections as graphs of semantically related text chunks. Nodes correspond to document segments, while edges represent lexical or semantic associations between them. During query processing, BrowseNet decomponses a question into directed acyclic graphs and then performs query-specific subgraph exploration to retrieve relevant information, combining both structural similarity (through graph traversal) and semantic similarity (through embeddings). The system is evaluated on multiple multi-hop QA datasets, showing competitive or superior performance to existing RAG, GraphRAG, and dense retrieval baselines.

Overall, this is an interesting and well-executed paper that tackles a relevant challenge in retrieval-based question answering: how to retrieve information that is indirectly connected through multiple associative steps. The approach is well-motivated, and the experimental setup is thorough. However, some of the claims, particularly regarding LLM independence and the nature of the “knowledge graph”, feel overstated or insufficiently substantiated.

**Strengths:**

1. Representing document chunks as graph nodes connected by lexical or semantic relations is a clear and intuitive way to capture associative structures in text.
2. The paper provides ablation studies and analyses of multiple components (e.g., graph generation, retrieval mechanisms, and QA results), which help the reader understand what drives performance.
3. The proposed method performs competitively, and in some cases surpasses, both dense retrieval and graph-based RAG baselines.
4. The paper is well-written and easy to follow, with good structure and figures that illustrate the key ideas effectively.

**Weaknesses:**

1. While the results are competitive, other claims such as BrowseNet minimizing dependence on LLMs are not really backed by how the method is desgined. It relies on LLMs at several stages, for creating the KG, embedding chunks, decomposing queries, etc. It is therefore not clear what the actual cost is in comparison with other methods.
2. An important source of cost is in retrieval for non-initiator nodes. This requires considering a total of $k^p$ chunks that need to be scored, increasing the cost of the method.
3. The query decomposition is based on a directed acyclic graph, which might lead to a limitation for queries that do involve cycles or cannot be expressed as a DAG. Whether this is a source of issues is not discussed.
4. Calling the constructed graph a “knowledge graph” seems somewhat misleading, since the edges primarily encode textual similarity or shared entity mentions, rather than semantic relations between well-defined concepts. It might be more accurate to refer to it as a semantic chunk graph or entity-linked similarity graph. Clarifying this terminology would prevent confusion for readers coming from the KG community.
5. Some improvements reported (e.g., Table 2, HotpotQA in Table 3) are small, and it’s unclear whether they are statistically significant. Including confidence intervals or significance tests (e.g., paired bootstrap) would increase confidence in the reported gains.

**Questions:**

1. Could you clarify in what sense the dependence on LLMs is reduced, or whether the key benefit lies more in structuring the outputs of LLMs rather than avoiding them altogether? Have you quantified the computational or monetary cost of these LLM-dependent components relative to baseline RAG methods?
2. The retrieval process for non-initiator nodes appears to involve scoring $k^p$ candidates. Could you provide more details about how this complexity behaves in practice?
3. How would BrowseNet handle queries that involve cycles, mutual dependencies, or other forms of recursive reasoning?
4. The isomorphic accuracy is an interesting way to measure the generated subgraphs. I assume that since the graphs are likely small, you used an exact algorithm for isomorphism check. Could you please elaborate on this?

---

> ### Author Response · Authors · 2025-11-22
> **Response to the reviewer 9MFQ**
>
> We thank the reviewer for their valuable feedback. Our response to the comments are as follows
>
> **Weakness-1 and Question-1:**
>
> We agree that BrowseNet uses LLMs for several components, including graph-of-chunks (KG) construction, chunk embedding, and query decomposition. In our paper, we use "LLMs" to specifically refer to models like Generative LLMs (Ex: GPT-class) systems that perform text generation. In contrast, NER (GliNER model) and embedding (NV-embed-V2 model) rely on smaller local models that do not incur the high computational or monetary cost associated with large generative LLM calls.
>
> **Clarifying our claim on reduced LLM dependence.**
> BrowseNet reduces LLM dependence. We would like to clarify that  BrowseNet does not claim to eliminate LLMs, but rather reduces reliance on generative LLMs during the retrieval phase. Existing multi-step RAG approaches (e.g., [1, 2, 3]) repeatedly invoke generative LLMs to iteratively decompose the query based on previously retrieved context. This can result in multiple expensive LLM calls per query. In contrast, BrowseNet uses the graph-of-chunks to guide retrieval and requires only *a single generative LLM call* for query decomposition. After this step, the retrieval procedure is entirely graph-guided and does not require additional generative LLM invocations. Although these iterative query-planning methods are well known, the HippoRAG paper has already shown them to be less effective than graph-based approaches; therefore, we did not include them as baselines but highlighted them briefly in the introduction for completeness.
>
> **Cost comparison.**
> Regarding computational cost, we have quantified the LLM-related overhead in Appendix A.7. Our analysis shows that BrowseNet achieves approximately 33× higher cost-efficiency than HippoRAG-2 (current SOTA model) while maintaining state-of-the-art retrieval performance. This improvement comes largely from avoiding repeated generative LLM calls.
>
> **Action:**
> We will clarify this distinction in the manuscript, explicitly state the meaning of minimizing LLM dependence during retrieval.
>
> **References:**
> 1) Harsh Trivedi, Niranjan Balasubramanian, Tushar Khot, and Ashish Sabharwal. Interleaving retrieval with chain-of-thought reasoning for knowledge-intensive multi-step questions. arXiv preprint arXiv:2212.10509, 2022a. (ACL 2023).
> 2) Yao Yao, Zuchao Li, and Hai Zhao. Beyond chain-of-thought, effective graph-of-thought reasoning in language models. arXiv preprint arXiv:2305.16582, 2023. (findings-NAACL 2024).
> 3) Yu Wang, Nedim Lipka, Ryan A Rossi, Alexa Siu, Ruiyi Zhang, and Tyler Derr. Knowledge graph prompting for multi-document question answering. In Proceedings of the AAAI Conference on Artificial Intelligence, volume 38, pp. 19206–19214, 2024.
>
>
> **Weakness-2 and Question-2:**
>
> Thank you for this important question. We clarify that while the k^p complexity is theoretically valid, practical factors might mitigate this concern.
> 1. Sparse Query Structures: Most evaluation datasets have shallow, sequential dependencies (p ≤ 4). Even with k=5 and p=2, we evaluate only 25 combinations.
> 2. Graph-Based Pruning: Candidate chunks are restricted to graph-of-chunk (knowledge graph) neighbors (Algorithm 1, lines 11-12), drastically reducing the effective search space versus corpus-wide evaluation.
> 3. Empirical Efficiency: BrowseNet's retrieval stage averages **1.19 seconds per query** (MuSiQue), with only **0.49 seconds additional overhead** compared to HippoRAG-2, while achieving substantially higher recall.
> 4. Beam Search Pruning: We retain only top-k scoring subgraphs, enabling early termination and further constraining computational cost.
>
> **Action:** We will add information about the complexity at the appendix section in the revised version

---

> ### Author Response · Authors · 2025-11-22
> **Response to the reviewer 9MFQ (continued)**
>
> **Weakness-3 and Question-3:**
>
> We appreciate the opportunity to clarify the design choice regarding DAG-based query decomposition. The following main reasons for the DAG-based query decomposition.
>
> Inherent nature of Question-Answering tasks: Multi-hop Question-Answering tasks should be fundamentally acyclic by design, as a well-posed question cannot logically require its own answer as a prerequisite. Doing so would create an infinite loop and violate the principle of well-founded reasoning. For example, if Q1 requires Q2 to be answered, and Q2 requires Q1, neither question can be answered, making the query semantically ill-defined. Hence, the DAG-based query decomposition is an appropriate choice for the Question-answering tasks.
> Dataset evidence: Furthermore,  examination across benchmarks (HotpotQA, 2WikiMQA, MuSiQue) shows that all gold decomposition chains are acyclic. No instances of cycles or mutual dependencies appear in naturally occurring multi-hop questions. This suggests that queries expressible as DAGs capture the full spectrum of practical QA scenarios.
>  By restricting decomposition to DAGs, we ensure: (i) *Termination guarantees*: Every subquestion has a well-defined base case, (ii) *Tractable reasoning*: No infinite loops or circular dependencies, (iii) *Semantic validity*: Decompositions remain faithful to the underlying question structure
>
> We acknowledge that if a user intentionally constructs a query with cyclic dependencies (e.g., "Answer Q1 if Q2 is true, and answer Q2 if Q1 is true"), BrowseNet would not handle it by design. However, such queries fall outside the scope of standard QA tasks and would require fundamentally different reasoning approaches (e.g., fixed-point computation or constraint satisfaction). For practical multi-hop QA, DAG-based decomposition is an appropriate choice.
>
> **Weakness-4**
>
> We recognize that a traditional knowledge graph involves entities as nodes and predicates as edges, while our graph is constructed over chunks of text, which suits retrieval-augmented generation tasks better. We initially followed prior work using the term "knowledge graph" [1][2] for similar graphs, but upon reflection, we agree the term could be clearer. We will revise the manuscript to replace "knowledge graph" with "graph of chunks" wherever appropriate
>
> **Action:**
> We will revise the manuscript to replace all instances of "knowledge graph" with "graph of chunks" to improve clarity and accuracy.
>
> **References:**
> 1) Wang, Y., Lipka, N., Rossi, R. A., Siu, A., Zhang, R., & Derr, T. (2024, March). Knowledge graph prompting for multi-document question answering. In Proceedings of the AAAI conference on artificial intelligence (Vol. 38, No. 17, pp. 19206-19214).
> 2) Yang, Z., Zhu, Z., & Zhu, J. (2025, April). CuriousLLM: Elevating multi-document question answering with llm-enhanced knowledge graph reasoning. In Proceedings of the 2025 Conference of the Nations of the Americas Chapter of the Association for Computational Linguistics: Human Language Technologies (Volume 3: Industry Track) (pp. 274-286).
>
> **Weakness-5**
>
> We agree that including confidence intervals and statistical significance testing would strengthen the validity of our reported improvements. In response, we conducted a paired bootstrap procedure with n = 10,000 resamples (α = 0.05, yielding 95% confidence intervals). We will report the corresponding confidence intervals for the retrieval results as shown in the table below.
>
> |    Method    | HotpotQA ||2WikiMQA||MuSiQue||
> |------|------|------|------|------|------|------|
> |  | R@2 [LB, UB] | R@5 [LB, UB] | R@2 [LB, UB] | R@5 [LB, UB] | R@2 [LB, UB] | R@5 [LB, UB] |
> | NV-embed-V2  | **83.95 [82.35, 85.55]**  | 95.64 [94.75, 96.50]   | 69.05 [67.52, 70.62]   |  76.72 [75.25, 78.23]| 53.31 [51.66, 54.98] | 69.86 [68.15, 71.56]  |
> | HippoRAG-2 |81.79 [80.10, 83.40] | 96.20 [95.30, 97.05]  | 75.45 [73.77, 77.07]   | 90.62 [89.53, 91.73]   | 53.63 [51.82, 55.45]   | 73.70 [72.02, 75.39]  |
> | BrowseNet  | 83.83 [82.25, 85.35] |**96.40 [95.55, 97.20]**  | **76.77 [75.13, 78.43]**   | **93.29 [92.38, 94.20]** |  **55.12 [53.44, 56.88]**  | **73.92 [72.25, 75.61]**   |
>
> In the table, LB refers to the lower bound of the confidence interval and UB refers to the upper bound of the confidence interval. Across all comparisons, we observe statistically significant improvements (p < 0.05) except in three cases: HotpotQA Recall@2 vs. NV-embed-V2, HotpotQA Recall@5 vs. HippoRAG-2 and MuSiQue Recall@5 vs. HippoRAG-2, where the differences are not statistically significant. However, even when retrieval differences are not significant, our method still achieves stronger question answering performance overall. This is because the subquery graph we generate guides the LLM more effectively toward the correct final answer.
>
> **Action:** We will include paired bootstrap-based confidence intervals in the revised manuscript to provide a more rigorous evaluation of BrowseNet’s performance gains.

---

> ### Author Response · Authors · 2025-11-22
> **Response to the reviewer 9MFQ (continued)**
>
> **Question-4**
>
> Thank you for your interest in our evaluation methodology. We clarify the implementation details of the isomorphism check below.
>
> **Algorithm Used:** We employ **NetworkX's `is_isomorphic()` function**, which implements the VF2 algorithm for graph isomorphism testing. This is an exact algorithm that provides deterministic results.
>
> **Computational Feasibility**
> As you correctly note, the query subgraphs in our evaluation datasets are small:
> - Average node count: 2-4 nodes (corresponding to 2-4 subquestions)
> - Maximum depth: 4 hops (MuSiQue dataset)
> - Typical structure: Linear or tree-like dependencies with minimal branching
>
> For graphs of this size, the VF2 algorithm runs in effectively constant time (< 1ms per comparison), making exact isomorphism checking tractable even across hundreds of test queries.
>
> **Scalability Note**
> For applications involving larger knowledge graphs or more complex queries, approximate graph matching algorithms could be substituted. However, for standard multi-hop QA benchmarks, exact isomorphism checking remains computationally efficient and provides the most rigorous evaluation metric.

---

> > ### Author Response · Authors · 2025-11-26
> > **Follow-up**
> >
> > Dear Reviewer,
> > I hope this message finds you well. As there is still about one week remaining in the discussion period, we would like to kindly follow up and ensure that our rebuttal has sufficiently addressed your concerns.
> > If there are any remaining questions or additional feedback you would like us to consider, please feel free to let us know. Your insights are highly valuable to us, and we would be glad to provide further clarification or revisions.
> >
> > Thank you again for your time and effort in reviewing our paper.
> >
> > Best regards,
> > The Authors

---

> > > ### Comment · Reviewer_9MFQ · 2025-11-26
> > >
> > > Thank you for your response. I believe your points address all my questions, with valuable additional results such as the sparse properties that result in better cost compared to the $k^p$ cost, the statistical tests, the clarification on the use of the term "knowledge graph", and the algorithm used for isomorphism check.
> > >
> > > I would be willing to increasing the score once the paper is updated with the actions described in the response.

---

> > > > ### Author Response · Authors · 2025-11-27
> > > > **Response to reviewer 9MFQ**
> > > >
> > > > Dear reviewer,
> > > >
> > > > Thanks for your response. We have uploaded the revised manuscript incorporating the suggested changes.
> > > >
> > > > The following are the changes made:
> > > > 1) We have modified the title, text, figure and the pseudo code to call the constructed graph as graph-of-chunks rather than knowledge graph.
> > > > 2) (lines 235-240): We have provided explanation on how the sparse properties that result in better cost compared to the cost incurred because of $k^p$ cost. Also the details of cost comparison are provided in Appendix A.6 and the same has been summarised in main text (lines 367-370).
> > > > 3) (Appendix: A-11): The implementation details and results for statistical significance of the retrieval results are provided. The reference to the same has been provided in the main text.
> > > > 4) (lines  297-301): We have provided details on the algorithm used for isomorphic accuracy.
> > > > 5) (lines 174-181): We have provided the reason for decomposing the query as directed acyclic graphs.
> > > >
> > > > Let us know if any further changes needs to be made.

---

### Official Review · Reviewer_oVjo · 2025-11-01

**Soundness:** 2
**Presentation:** 2
**Contribution:** 2
**Rating:** 4
**Confidence:** 4

**Summary:**

This paper proposes BrowseNet, a knowledge graph–based associative memory framework for multi-hop contextual information retrieval. Unlike traditional RAG systems that rely solely on semantic similarity, BrowseNet constructs a knowledge graph where document chunks are nodes enriched with embeddings and lexical entity links form edges. For each query, BrowseNet decomposes it into sub-queries and performs structured graph traversal to retrieve relevant subgraphs that better capture reasoning chains. Experiments on HotpotQA, 2WikiMQA, and MuSiQue show state-of-the-art performance in recall and exact match compared to dense and graph-based RAG baselines, while reducing LLM interaction cost through more efficient retrieval.

**Strengths:**

Proposes a knowledge graph–based traversal approach that decouples queries for more accurate and context-aware retrieval, achieving competitive results in both passage retrieval and answer generation.

GraphRAG is an important domain for facts-required questions answer.

**Weaknesses:**

The core idea focuses on context retrieval, but the novelty is limited since query decomposition and graph-based iterative retrieval have been explored previously; comparison to similar methods (e.g., SiReRAG, ArchRAG, GraphRAG) is missing.


Zhang, N., Choubey, P. K., Fabbri, A., Bernadett-Shapiro, G., Zhang, R., Mitra, P., ... & Wu, C. S. (2024). Sirerag: Indexing similar and related information for multihop reasoning. arXiv preprint arXiv:2412.06206.
Wang, S., Fang, Y., Zhou, Y., Liu, X., & Ma, Y. (2025). ArchRAG: Attributed Community-based Hierarchical Retrieval-Augmented Generation. arXiv preprint arXiv:2502.09891.
Han, H., Wang, Y., Shomer, H., Guo, K., Ding, J., Lei, Y., ... & Tang, J. (2024). Retrieval-augmented generation with graphs (graphrag). arXiv preprint arXiv:2501.00309.

Scalability to large, real-world corpora and real-time use cases is not clearly addressed, as experiments rely on controlled corpora with gold evidence and distractors.

Fairness of comparison is unclear, particularly whether the same backbone models (generation, NER, embeddings) were used across baselines, which may confound reported improvements.

The current framework is optimized for structured multi-hop reasoning and may not directly generalize to open-domain retrieval or tasks with unstructured context dependencies.

The approach depends on LLM-based query decomposition, which may introduce structural or semantic errors when sub-queries are misgenerated, leading to cascading retrieval failures.

**Questions:**

See weakness

---

> ### Author Response · Authors · 2025-11-22
> **Response to the reviewer oVjo**
>
> We thank the reviewer for the valuable feedback. Our response to the comments are as follows
>
> **Weakness-1:**
>
> We agree these approaches have been studied before; however, the novelty of BrowseNet lies in its unified pipeline that integrally combines query decomposition, graph-of-chunks construction, and efficient one-shot retrieval, which to our knowledge is not previously realized. In response to the reviewer’s suggestion, we have benchmarked BrowseNet against related methods SiReRAG and GraphRAG. Unfortunately, the ArchRAG code is not publicly available, and the anonymous access link appears expired. We have contacted the authors and will include comparisons once the code becomes accessible.
>
> Below are comparison results on HotpotQA, 2WikiMQA, and MuSiQue datasets using the gpt-4o-mini answer generation model with top-5 chunks provided:
>
> |    Method (K=5)  | HotpotQA ||2WikiMQA||MuSiQue||Average||
> |------|------|------|------|------|------|------|------|------|
> |  | EM | F1 | EM | F1 | EM | F1 | EM | F1 |
> | Browsenet (gpt-4o-mini) | **62.20**  | **77.69**   | **63.90**   | **74.50**  | **41.60**  | **54.08**    | **55.90**   |  **68.76**  |
> | SiReRAG (gpt-4o-mini)  | 48.30  | 63.17  | 41.30   | 48.05   | 26.00   | 39.59  | 38.53  |50.27   |
> | GraphRAG (gpt-4o-mini) | 51.40   | 67.60   | 45.70   | 61.00   | 27.00   | 42.00   | 41.37   | 56.87   |
>
> In the main paper of SiReRAG, top-20 chunks were provided, hence for comparison with BrowseNet, we have implimented again with top-20 chunks and the results for that are shown below.
>
> |    Method (K=20)   | HotpotQA ||2WikiMQA||MuSiQue||Average||
> |------|------|------|------|------|------|------|------|------|
> |  | EM | F1 | EM | F1 | EM | F1 | EM | F1 |
> | Browsenet (gpt-4o-mini) | **62.20**  | **78.36**   | **68.20**   | **78.47**  | **44.50**  | **57.77**    | **58.3**   | **71.53**   |
> | SiReRAG (gpt-4o-mini)  | 52.80  | 69.17  | 46.10   | 54.82   | 29.80   | 44.96  | 42.90  |  56.32  |
>
> As seen from both the Tables, BrowseNet achieves superior performance, establishing itself as the state-of-the-art in this retrieve-then-read framework setting. As both of these methods do not follow retrieve-then-read paradigm, we did not include the results for retrieval.
>
> **Action:** We will incorporate these comparison results with SiReRAG and GraphRAG (for K=5 ) into the paper’s result section.
>
> **Weakness-2:**
>
> To better reflect real-world use cases, we have modified the benchmark datasets (HotpotQA, MuSiQue, and 2WikiMultiHopQA) by including all passages from other questions as candidate distractors. As shown in Table-1, the number of nodes reflects the number of passages that are given as candidate passages for all the questions in the benchmark dataset. The number of passages for the benchmarks are 9,221, 6,119 and 11,656 respectively for HotpotQA, 2WikiMQA and Musique. This effectively enlarges the candidate corpus, simulating a more realistic retrieval setting where numerous irrelevant documents must be filtered.
>
> **Action**: We will clearly describe this benchmark modification in the manuscript to demonstrate BrowseNet’s applicability and robustness in larger, more challenging retrieval scenarios.
>
> **Weakness-3:**
>
> Thank you for raising the concern regarding the fairness of comparison. We clarify that simple baselines and dense retrievers do not utilize LLMs. For comparable methods such as RAPTOR, HippoRAG, HippoRAG-2, LightRAG, GraphRAG and SiReRAG, we have consistently implemented gpt-4o-mini for indexing, retrieval, and question answering to ensure a fair evaluation. Specifically, HippoRAG-2 requires an embedding model, for which we used NV-Embed-v2 the same embedding model employed by BrowseNet. Furthermore, as reported in the main results and ablation studies, BrowseNet demonstrates superior performance using NV-Embed-v2 and maintains robustness regardless of the choice of NER models (GliNER, gpt-4o, Claude-3.7-sonnet), subquery decomposition models (gpt-4o, Deepseek Reasoner, Claude-3.7-sonnet, gpt-4o-mini), or generation models (gpt-4o-mini, gpt-3.5-turbo, gpt-4.1-mini, Deepseek-chat-v3, Gemini-2.0-flash).
>
> **Action**: We will clarify these backbone model settings across baselines and our method to emphasize fairness and robustness of the comparison.

---

> ### Author Response · Authors · 2025-11-22
> **Response to reviewer oVjo (continued)**
>
> **Weakness-4:**
>
> BrowseNet is designed to be robust to varying query structures. When a query cannot be decomposed into subqueries, retrieval relies solely on initiator nodes, effectively performing a semantic search over the entire corpus. Additionally, during retrieval for each subquery, similarity scores are also computed between chunks and the original multi-hop query, allowing the system to maintain relevance to the full query context. Also, BrowseNet only requires keyword prediction for constructing the graph-of-chunks, hence it creates structured context from the unstructured data and works for any multi-hop reasoning setting. Exploring how performance improves by leveraging such structural preprocessing remains an interesting direction for future work.
>
> **Action:** We will include this point in the final revised version of the manuscript
>
> **Weakness-5:**
>
> The approach depends on LLM-based query decomposition, which may introduce structural or semantic errors when sub-queries are misgenerated, leading to cascading retrieval failures.
>
> Query decomposition is a critical element of our approach and hence we have performed a detailed error analysis on the subquery generation on answer generation. We have summarised the error analysis in main text and presented it in a detailed manner in Appendix A.10.

---

> ### Author Response · Authors · 2025-11-26
> **Follow up**
>
> Dear Reviewer,
> I hope this message finds you well. As there is still about one week remaining in the discussion period, we would like to kindly follow up and ensure that our rebuttal has sufficiently addressed your concerns.
> If there are any remaining questions or additional feedback you would like us to consider, please feel free to let us know. Your insights are highly valuable to us, and we would be glad to provide further clarification or revisions.
>
> Thank you again for your time and effort in reviewing our paper.
>
> Best regards,
> The Authors

---

### Official Review · Reviewer_Jw6M · 2025-11-01

**Soundness:** 3
**Presentation:** 3
**Contribution:** 3
**Rating:** 6
**Confidence:** 4

**Summary:**

This paper proposed a method for gathering context for retrieval-augmented generation (RAG): first process the candidate documents into a form of graph (where nodes are chunks and edges exist when there are shared entities), and explore according to a query-specific subgraph generated by LLMs. The method achieves state-of-the-art performance across a wide range of question answering tasks that require multi-hop reasoning.

**Strengths:**

- The preprocessing of the corpus is intuitive and based on entities. The idea is widely applicable to other NLP tasks
 - Evaluation is done at all stages of the RAG pipeline: at the construction of the graph, subgraph planning, retrieval, and final answer generation. The evaluations presented are convincing.

**Weaknesses:**

- The graph constructed probably should not be called a "knowledge graph" -- usually in the context of information extraction a KG has entities as nodes and predicates as edges. The proposal here is a graph on chunks of text (which I agree that is a better format for downstream RAG than a traditional 3-tuple KG)
 - The context is gathered with one shot -- maybe one could execute retrieval multiple times as the LLM executes the query subgraph it generates? As in a beam-search like process, the retrieved context could evolve as the LLM tries to solve the question (as in the style of https://aclanthology.org/2023.acl-long.557/)?
 - A snippet subgraph of the constructed graph would be illuminating-- I suggest the authors present one in the final version.

**Questions:**

- L195: $cos$ has a single argument: to denote cosine similarity, please write $\cos \angle (x, y)$.
- L202-L220: Please clarify this search process on how this is related to beam search.

---

> ### Author Response · Authors · 2025-11-22
> **Reply to reviewer Jw6M**
>
> We thank the reviewer for the valuable feedback. Our response to the comments are as follows
>
> **Weakness-1:**
>
> We recognize that a traditional knowledge graph involves entities as nodes and predicates as edges, while our graph is constructed over chunks of text, which suits retrieval-augmented generation tasks better. We initially followed prior work using the term "knowledge graph" [1][2], but upon reflection, we agree the term could be clearer. We will revise the manuscript to replace "knowledge graph" with "graph of chunks" wherever appropriate
>
> **Action:**
> We will revise the manuscript to replace all instances of "knowledge graph" with "graph of chunks" to improve clarity and accuracy.
>
> **References:**
> 1) Wang, Y., Lipka, N., Rossi, R. A., Siu, A., Zhang, R., & Derr, T. (2024, March). Knowledge graph prompting for multi-document question answering. In Proceedings of the AAAI conference on artificial intelligence (Vol. 38, No. 17, pp. 19206-19214).
> 2) Yang, Z., Zhu, Z., & Zhu, J. (2025, April). CuriousLLM: Elevating multi-document question answering with llm-enhanced knowledge graph reasoning. In Proceedings of the 2025 Conference of the Nations of the Americas Chapter of the Association for Computational Linguistics: Human Language Technologies (Volume 3: Industry Track) (pp. 274-286).
>
> **Weakness-2:**
>
> The core idea of BrowseNet is to minimize the number of LLM interactions during retrieval by leveraging a graph-of-chunks structure. Unlike IRCoT, which interleaves retrieval and reasoning by updating the query at each step over the entire corpus, BrowseNet restricts the retrieval candidates to neighbors in the graph. This focus reduces retrieval complexity and enables a one-shot retrieval approach by splitting the query, thus limiting LLM interaction to a single retrieval step. In contrast, IRCoT’s number of LLM interactions depends on query complexity and can be higher.
>
> **Action:** We will clarify this distinction in the manuscript to highlight BrowseNet’s aim of efficient retrieval via one-shot interaction and graph-based candidate reduction, contrasting it with IRCoT’s iterative retrieval process.
>
> **Weakness-3:**
>
> **Action:**
> We will include an illustrative snippet subgraph in the revised version of the paper, demonstrating the structure of the graph-of-chunks and how it relates to the decomposed query-subgraph. This visual aid will clarify the relationships between chunks and exemplify the retrieval process
>
> **Questions-1:**
>
> Thanks for pointing that out, we will update the notation used for the cosine similarity
>
> **Questions-2:**
>
> In BrowseNet, the retrieval over the knowledge graph during multi-hop questions follows a principled graph traversal guided by the decomposed query-subgraph. For non-initiator nodes in the query-subgraph, we consider all combinations of candidates retrieved from predecessor nodes and gather their neighbors as the candidate corpus for the next retrieval step. From this expanded candidate set, we score chunks semantically relative to the current subquery and select the top-k scoring subgraphs. This scoring and selection over multiple candidate subgraphs resembles a beam search that maintains a fixed-width set of best subgraph paths as retrieval proceeds along the query-subgraph in topological order. Thus, while the traversal is not beam search in the classic sequence generation sense, it analogously explores multiple hypotheses (candidate subgraphs) at each step, pruning them according to a scoring function to efficiently focus retrieval on promising reasoning paths.
>
> **Action:** We will clarify this analogy to beam search in the paper and explicitly describe the candidate combination, scoring, and pruning process during subgraph retrieval with a figure in appendix to improve reader understanding.

---

> ### Author Response · Authors · 2025-11-26
> **Follow up**
>
> Dear Reviewer,
> I hope this message finds you well. As there is still about one week remaining in the discussion period, we would like to kindly follow up and ensure that our rebuttal has sufficiently addressed your concerns.
> If there are any remaining questions or additional feedback you would like us to consider, please feel free to let us know. Your insights are highly valuable to us, and we would be glad to provide further clarification or revisions.
>
> Thank you again for your time and effort in reviewing our paper.
>
> Best regards,
> The Authors

---

### Comment · Area_Chair_1uNh · 2025-11-26

Dear Reviewers,

The authors have posted a rebuttal. Please engage. This is a part of your reviewing responsibilities. It is important to engage early so that the authors get a fair chance to respond to any follow-up queries.

best,

AC

---

### Author Response · Authors · 2025-12-01
**Summary of contribution and changes made in the manuscript**

We would like to thank reviewers for your time and attention for evaluating our work and discussions, and for appreciating our contribution, BrowseNet, in the areas of multi-hop question-answering using Graph-based RAG.

**Contribution and novelty:**
The main contribution of BrowseNet is to exploit the query’s structural and semantic features in traversing the graph-of-chunks, such that Retrieval is achieved with a single LLM interaction, thereby reducing cost and latency. The novelty of BrowseNet stems from guiding the multi-hop retrieval procedure based on the semantic closeness to the query and traversing the graph-of-chunks based on lexical connections between the chunks.  This novelty of BrowseNet leads to achieving state-of-the-art (SOTA) performance in multi-hop question answering, outperforming both dense retrievers and graph-based RAG systems on the benchmark datasets.

We have incorporated the suggestions of the reviewers to strengthen our work and have made changes to the revised manuscript (highlighted in red color), incorporating the changes suggested by the reviewers, along with the additional experiments conducted in response to their comments.

**The summary of the changes in the manuscript is as follows.**

1) Revised the title, text, figure, and the pseudo code to call the constructed graph a graph-of-chunks rather than a knowledge graph. (Response to Reviewers: Jw6M, 9MFQ, Dpwg)
2) An explanation of how sparsity in the proposed graph results in lower computational cost compared to the $k^p$ cost is added (refer to lines 237-242). Detailed cost comparisons are provided in Appendix A.6, with a summary included in the main text. (refer to lines 367-370). (Response to Reviewer 9MFQ)
3) The implementation details and statistical significance results for the retrieval performance in the Appendix: A-11, Table 15, are included.  Corresponding references have been added to the main text. (Response to Reviewer 9MFQ)
5) Details of the algorithm used to compute isomorphic accuracy are provided in lines 308-311. (Response to Reviewer 9MFQ)
6) The motivation and reasons for decomposing queries into directed acyclic graphs are clarified and explained  (refer to lines 174-179). (Response to Reviewer  9MFQ)
7) A snippet subgraph of the graph-of-chunks derived from the 2WikiMultiHopQA dataset is added in Figure-2. (Response to Reviewer Jw6M)
8) The relationship between the proposed graph traversal strategy and the beam search method is clarified and explained in lines 232-235 and  Figure 3. (Response to Reviewer Jw6M)
9) Updated and unified the notation for cosine similarity in Equation (1) and Algorithm 1. (Response to Reviewer Jw6M)
10) The question answering results for the new method, SiReRAG are added in Table 3. (Response to Reviewers oVjo, Dpwg)
11) We have clarified how benchmark datasets were modified to include distractor passages from other questions, thereby better reflecting real-world scenarios (refer to lines 268-272). (Response to Reviewers oVjo, Dpwg)
12) The queries on the fairness of the experimental comparison, specifically with respect to the use of generative LLMs across all baselines, are explained and clarified (refer to lines 283-289). (Response to Reviewer oVjo)
13) How BrowseNet falls back to semantic search when a query cannot be decomposed is clarified. (refer to lines 207-209) (Response to Reviewer oVjo)

During the discussion period, two Reviewers who engaged during the discussion period (9MFQ, Dpwg) have responded positively after reading our responses and the action taken to improve the manuscript before the suspension of the discussion phase of ICLR 2026.

We believe that the revised version addresses all the concerns raised and strengthens the manuscript for publication.

---

### Meta-Review · Area_Chair_hpQc · 2025-12-07

**Summary:**

The paper proposes BrowseNet, a retrieval-augmented generation (RAG) framework for multi-hop question answering. The core contribution involves constructing a "graph-of-chunks" (nodes as text chunks, edges based on lexical/semantic overlap) and utilizing an LLM-based query decomposition to guide a one-shot retrieval process. This approach aims to balance the structural benefits of knowledge graphs with the flexibility of dense retrieval, reducing the latency and cost associated with iterative retrieval methods.

The decision to accept is based on the paper's strong empirical performance on standard benchmarks (HotpotQA, 2WikiMultiHopQA, MuSiQue), where it demonstrates state-of-the-art results compared to strong baselines. During the rebuttal, the authors significantly strengthened the manuscript by adding requested baselines (SiReRAG, GraphRAG), clarifying the computational cost/efficiency, and refining the terminology. While some reviewers questioned the novelty of individual components, the unified pipeline's effectiveness and the robust experimental validation justify acceptance.

**Reviewer Concerns:**

Addressed by Rebuttal:

1. Terminology ("Knowledge Graph"): Reviewers Jw6M, 9MFQ, and Dpwg correctly pointed out that the proposed structure is a graph of text chunks, not a traditional entity-predicate Knowledge Graph. The authors agreed to rename the structure "Graph of Chunks" throughout the paper.

2. Missing Baselines: Reviewers oVjo and Dpwg requested comparisons against recent methods like SiReRAG and GraphRAG. The authors provided these comparisons in the rebuttal, showing BrowseNet's superior performance (e.g., significantly higher EM/F1 on HotpotQA compared to SiReRAG).

3. Statistical Significance: Reviewer 9MFQ requested confidence intervals. The authors added paired bootstrap significance tests, confirming that the improvements are statistically significant.

4. Cost and LLM Dependence: Reviewer 9MFQ questioned the claim of reduced LLM dependence. The authors clarified that the reduction refers specifically to generative LLM calls during the retrieval loop (using cheaper embedding/NER models instead), and provided a cost analysis showing the method is ~33x more cost-efficient than HippoRAG-2.

5. Fairness of Comparison: Reviewer oVjo questioned if backbone models were consistent. The authors confirmed that all baselines used the same generation (GPT-4o-mini) and embedding models where applicable, ensuring fair comparison.

Outstanding / Unresolved:

1. Novelty of Components: Reviewers oVjo and Dpwg noted that query decomposition and graph construction are standard techniques. While the authors argue that the novelty lies in the specific unified pipeline and the query-guided traversal strategy, Reviewer Dpwg remained only partially convinced regarding novelty, though they acknowledged the method's effectiveness.

**Reviewer Scores:**

1. Reviewer Jw6M (Score: 6 -> 6): The reviewer was already positive.

2. Reviewer oVjo (Score: 4 -> 4): This reviewer did not engage in the final discussion.

3. Reviewer 9MFQ (Score: 6 -> 8): The reviewer explicitly stated, "I would be willing to increasing the score once the paper is updated," after the authors provided the statistical tests and cost analysis.

4. Reviewer Dpwg (Score: 2 -> 4): The reviewer explicitly stated, "I will increase my score," acknowledging that the responses regarding baselines and terminology were satisfactory, despite lingering reservations about novelty.

---

### Decision · Program_Chairs · 2026-01-26

Accept (Poster)